# An at-leg pellet and associated *Penicillium* sp. provide multiple protections to mealybugs
Zicheng Li [1,3], Haojie Tong [1,2,3], Meihong Ni[1], Yiran Zheng[1], Xinyi Yang[1], Yumei Tan[1], Zihao Li [1] & Mingxing Jiang [1] ✉

Beneficial fungi are well known for their contribution to insects' adaptation to diverse habitats. However, where insect-associated fungi reside and the underlying mechanisms of insect-fungi interaction are not well understood. Here, we show a pellet-like structure on the legs of mealybugs, a group of economically important insect pests. This at-leg pellet, formed by mealybugs feeding on tomato but not by those on cotton, potato, or eggplant, originates jointly from host secretions and mealybug waxy filaments. A fungal strain, *Penicillium citrinum*, is present in the pellets and it colonizes honeydew. *P. citrinum* can inhibit mealybug fungal pathogens and is highly competitive in honeydew. Compounds within the pellets also have inhibitory activity against mealybug pathogens. Further bioassays suggest that at-leg pellets can improve the survival rate of *Phenacoccus solenopsis* under pathogen pressure, increase their sucking frequency, and decrease the defense response of host plants. Our study presents evidences on how a fungi-associated at-leg pellet provides multiple protections for mealybugs through suppressing pathogens and host defense, providing new insights into complex insect × fungi × plant interactions and their coevolution.

Symbiotic fungi often play pivotal roles in animals by providing nutrients, defending hosts against natural enemies, degrading polymeric materials (e.g., wood), and regulating behavior[1]. In insects, the most diverse class of animals in the world, beneficial fungi are well known for their contribution to the adaptation of their hosts to diverse habitats[1]. The associated fungi can be housed in insect tissues or cells, e.g., *Yarrowia* yeasts in the gut of necrophagous beetles[2] and yeast-like symbionts in the bacteriocytes of *Nilaparvata lugens*[3]; inside specific organs on insect integument, such as the fungal symbionts-storing mycangium of ambrosia beetles[4]; or in the environment surrounding the insects, such as the fungi in termite nests[5]. Beneficial fungi often benefit associated insects indirectly, e.g., by interacting with and subsequently suppressing their natural enemies including pathogens[1], predators[6], and parasitoids[7], secreting enzymes to degrade the adverse secondary metabolites contained in host food[8], or by modulating microbial communities to create an environment suppressive to pathogens[9], depending on the specific insect-fungi associations. In each of these cases, the associations are thought to have arisen through long-term coevolution[10].

Including associations with symbiotic fungi, insects have adopted various approaches to adapt to environments, depending on their taxa. Mealybugs (Order Hemiptera, Family Pseudococcidae), as a group of insects containing at least 2000 species[11,12], are widely distributed throughout the world, have a broad host range, and are economically important plant pests. Some mealybug species, such as the cotton mealybug, *Phenacoccus solenopsis*, damage crops and landscape trees, causing losses in yield or ornamental value[12]. Mealybugs, though wingless except for adult males, lead a sedentary lifestyle and have adapted to various environments through strategies like coating their bodies with a thick layer of powdery wax and utilizing nutrients from bacterial endosymbiosis[11,13].

Recently, we discovered peculiar pellets that "wrapped" the pretarsi and tarsi of *P. solenopsis* while feeding on tomato plants (*Solanum lycopersicum* L.), often found on multiple legs of the same insect. This discovery intrigued us, as substances attached to the legs of other insects, such as leafhoppers and aphids feeding on *Solanum* plants, can prove fatal by hindering movement and feeding[14,15]. However, mealybugs with the pellets at leg live very well (personal observation). Moreover, such masses similar to

[1]Institute of Insect Sciences, Key Laboratory of Biology of Crop Pathogens and Insects of Zhejiang Province, Key Laboratory of Molecular Biology of Crop Pathogens and Insects, Ministry of Agriculture and Rural Affairs of the People's Republic of China, State Key Laboratory of Rice Biology, Zhejiang University, Hangzhou, China. [2]College of Life Sciences, China Jiliang University, Hangzhou, China. [3]These authors contributed equally: Zicheng Li, Haojie Tong. ✉e-mail: mxjiang@zju.edu.cn

the pellets of mealybugs were ever reported in Colorado potato beetle larvae, (*Leptinotarsa decemlineata*), where encasement of tarsi by materials from host potato *Solanum polyadenium* trichomes led to loss of grip and falling off[46]. Thus, pellets on legs may be unique to certain insect taxa using *Solanum* plants as hosts, historically regarded as harmful. This prompts exploration into why *P. solenopsis* retains such pellets and whether they confer any benefits.

Furthermore, examination of the pellet surface revealed numerous spherical structures resembling fungal spores, sparking further interest. Confirmation of these structures as fungal spores would suggest a potential relationship between mealybugs and fungi. The origin of these spores is of particular interest, given that secretions from tomato glandular trichomes typically possess antimicrobial properties[17], making them inhospitable to most fungi. Therefore, investigating the type, source, and function of these fungi in symbiotic relationships with mealybugs presents an intriguing avenue for research.

In order to explore how mealybugs may use any associated beneficial fungi to adapt to their environment, here we report a novel insect-microbe association in mealybugs. We conducted several experiments to investigate the effects of newly discovered at-leg pellets on *P. solenopsis*, as well as to explore the lifecycle of an associated symbiotic fungus (*Penicillium citrinum*). Our aims were to clarify the role of the at-leg pellets and associated *Penicillium* sp. in defending *P. solenopsis* and plant hosts (*S. lycopersicum*) from pathogens, regulating mealybug behavior and host defenses, and colonizing new insect hosts.

## Results
### Characterization and source of pellets on mealybug legs
As revealed by stereo and scanning electron microscopy, the pellets at the leg of cotton mealybugs (*P. solenopsis*) feeding on tomato plants under both phytotron and field conditions were solid, irregularly shaped structures that "wrapped" the entire tarsus and pretarsus and part of the tibia (Fig. 1a).

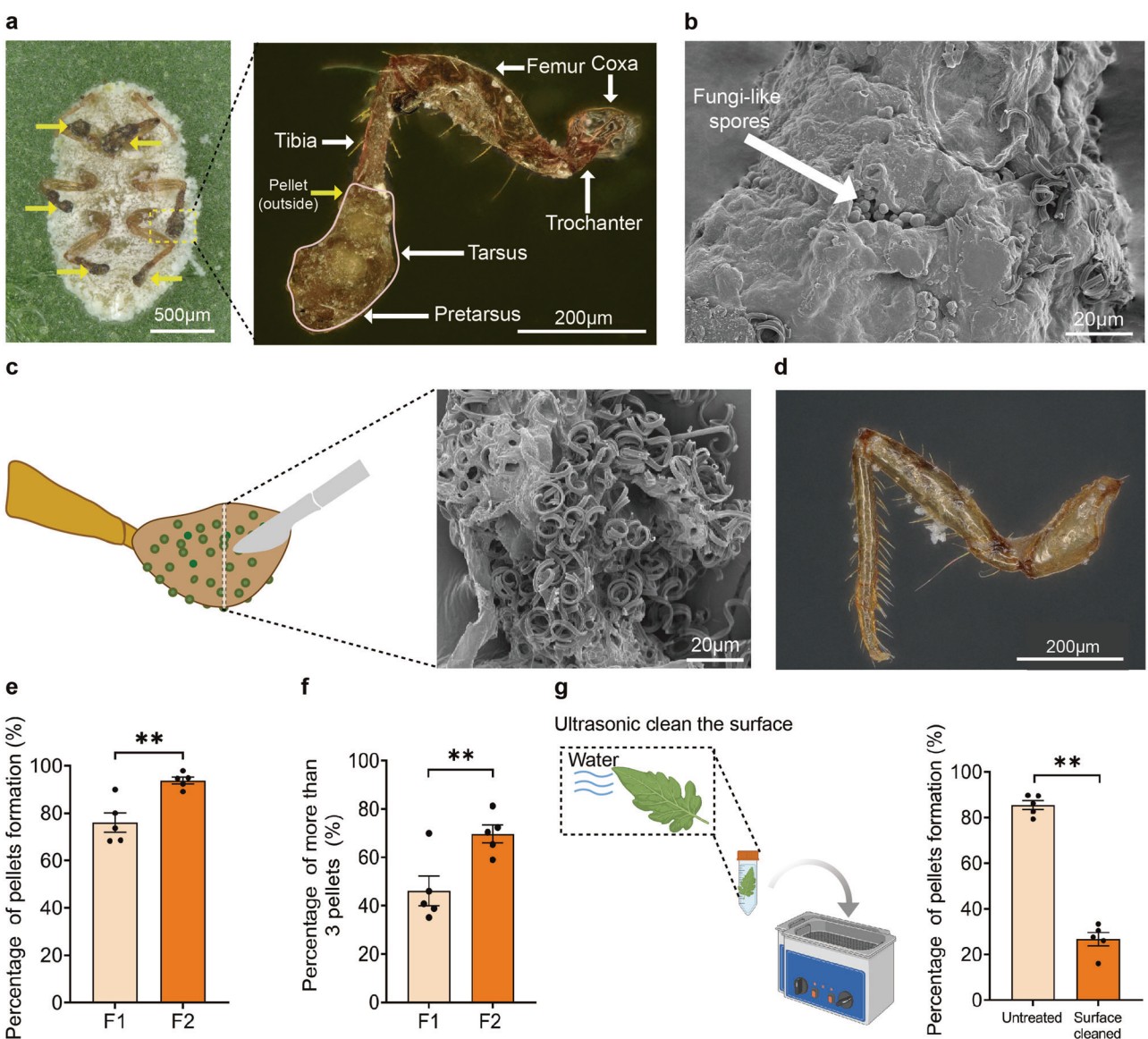

**Fig. 1 | Morphology, structure, and origin of the pellets at legs of the cotton mealybug, Phenacoccus solenopsis. a** A female adult bearing the pellets (indicated with yellow arrows; left), which wraps the entire tarsus and pretarsus and part of the tibia (right). **b** Surface structure of pellets. The spheres indicated with white arrow are fungi-like spores. **c** Internal structure of a pellet. **d** Leg of a newly emerged female adult without pellet. **e** Percentage of mealybugs forming pellets in two successive generations, F1 (20–91 insects per replicate) and F2 (48–147 insects per replicate).

**f** Percentage of the mealybugs with more than three pellets in F1 and F2 generation. **g** Percentage of mealybugs forming pellets on intact (i.e. untreated) and ultrasonic surface cleaned tomato leaves (19–29 and 23–30 insects in per replicate, respectively). **e**–**g** Bars represent the mean ± SE while points represent each biological replicate; the number of asterisk represent the *p* value (*$P < 0.05$; **$P < 0.01$) as determined by a Student's *t* test.

Waxy filaments and spheres were found on the pellet surface (Fig. 1b, c). The pellets were lost at molting (Fig. 1d), but they could be reattained by *P. solenopsis* around 12 h on tomato. As mealybugs progressed through their life cycle on tomato plants, the density of glandular trichomes and their secretions on plants increased[18], leading to a notable rise in both the percentage and quantity of pellets observed (Fig. 1e, f; $P < 0.05$). Such pellets were also found in the solanum mealybug (*Phenacoccus solani*) and the papaya mealybug (*Paracoccus marginatus*) following feeding on tomato (Supplementary Fig. 1a).

As glandular trichomes are frequently present on Solanaceae plants, particularly those of *Solanum*, taken by plants to defend herbivores[19,20], we speculated that the pellets observed on mealybug legs probably originated from the secretions of tomato plants. To test this, we compared the percentage of *P. solenopsis* female adults that attained pellets at the leg on intact tomato leaves with the percentage of those on the tomato leaves with their secretions removed previously. Our results showed that pellets were formed in 85.4 ± 1.8% (Fig. 1g) of individuals on intact tomato leaves; in contrast, the percentage was only 26.7 ± 2.6% on the secretions-removed leaves, significantly lower than the former ($P < 0.001$). We also examined the situation on cotton (*Gossypium hirsutum*), which exhibits minimal plant secretions on its leaves or stems, as well as on potato (*Solanum tuberosum*) and eggplant (*Solanum melongena*), both of which have fewer types of trichomes compared to tomato plants[19,21–23]. No pellets were formed on each of these plants (Supplementary Fig. 1b-d). Therefore, secretions of glandular trichomes are the source of the pellets at the legs of mealybugs feeding on tomato.

### Only *P. citrinum* was detected in mealybug at-leg pellets

As the spheres on pellet surfaces (Fig. 1b) have a diameter similar to fungal spores (nearly 4.0 μm[24]), there might be certain fungi in the pellets. Inoculating legs with pellets from field-collected and laboratory-reared mealybugs onto potato dextrose agar (PDA) plates, we obtained fungal isolates that were similar in hyphal and spore morphology, without other kinds of isolates. This result was generated consistently while inoculating legs of mealybugs sampled at different times. The isolates were identified as *P. citrinum* (GenBank ID: OR647500), as indicated by both morphological features and phylogenetic analysis of their internal transcribed spacer (ITS) region (Fig. 2a, b and Supplementary Fig. 2). Its ITS 543-bp fragment shows a 100% similarity with that of accession KX664347[25]. Conversely, when we inoculated pellet-free legs of mealybugs feeding on surface-cleaned tomato leaves onto PDA plates, we isolated only 1–2 strains each from two (6.7%) of the sampled mealybugs (totaling 30 individuals), with none obtained from other individuals (Fig. 2c). These isolates were identified as *Aspergillus* and *Cercospora* (GenBank ID: PP338193 and PP338195, respectively), indicating that the *P. citrinum* identified originated from pellets rather than the legs carrying them. When we inoculated legs with pellets from field-sampled mealybugs on plates of Czapek dox agar (CzA), yeast extract peptone dextrose (YPD), and Sabouraud dextrose agar (SDA), only *P. citrinum* was cultured (Supplementary Fig. 3a, c, e), displaying an identical ITS sequence to the previously described *P. citrinum*. No fungi were cultured from legs lacking pellets on these plates (Supplementary Fig. 3b, d, f).

### *P. citrinum* exhibited antimicrobial activity against pathogens

*Penicillium* spp. have broad inhibitory properties[26] and those associated with insects typically colonize plants, providing antimicrobial-based protection to both insects and plants[27]. We tested whether the *P. citrinum* identified from pellets plays such roles for mealybugs. The results showed that the *P. citrinum* fermentation (PCF) products (Fig. 2d) could inhibit *Akanthomyces lecanii* (Fig. 2e) and *Beauveria bassiana* (Fig. 2f), two fungal pathogens of mealybugs. PCF products also inhibited *Pseudomonas syringae* pv. tomato (Later written as *P. syringae*) to certain degree, a bacterial pathogen of tomato (Fig. 2g), but did not inhibit *Botrytis cinerea*, a fungal pathogen of tomato (Fig. 2h). In the control experiment, no antimicrobial activity was detected in the assay using products extracted from rice (RE, containing PDA liquid media), which were utilized in preparing PCF

products (Supplementary Fig. 4). Thus, the antimicrobial activity observed in the PCF products was attributed to *P. citrinum* rather than RE.

To determine which compounds contribute to the antimicrobial activity of *P. citrinum*, we investigated the kinds of compounds in PCF products and RE. Two dominant components were identified, butylparaben (BP) and citrinin, in the PCF products (Supplementary Table 1), and none of them were found in RE (Fig. 2i, j). BP inhibited *A. lecanii* (Fig. 2k), which is accordant with a previous report that BP has antimicrobial properties[28], but it did not inhibit *P. syringae* (Fig. 2l). In contrast, citrinin did not inhibit *A. lecanii* and *P. syringae* (Fig. 2m, n).

### Colonization and proliferation of *P. citrinum* in honeydew and horizontal transmission to mealybugs

The presence of *P. citrinum* spores in pellets (Fig. 1b) implies that this fungus must exist and be able to reproduce somewhere accessible to mealybugs. Honeydew produced by *P. solenopsis* might be such a place, because, high in sugar concentration, honeydew can be selectively colonized by certain plants- or insects-associated fungi[29,30]. Our observation showed that, under greenhouse conditions, *P. citrinum* had already colonized at 7 d in the honeydew on tomato stems that were previously smeared with sterilized honeydew and released with pellets-harboring mealybugs (Fig. 3a), and at 14 d, *P. citrinum* in the honeydew occurred as yellow or green colonies on plants (Fig. 3b–d). In comparison, no *P. citrinum* colonies were found in the honeydew smeared to tomato stems free of mealybugs (Fig. 3e, f). Moreover, *P. citrinum*, together with *Cercospora*, could be isolated from the honeydew collected from tomato plants naturally infested by *P. solenopsis* under field conditions (Fig. 3g). The *Cercospora* isolate was previously reported as a pathogen of tomato[31], and it was found in the honeydew collected from our greenhouse (Fig. 3h).

In the in vitro assay where pellets-free mealybugs were released onto honeydew inoculated with *P. citrinum*, 45.2 ± 1.3% of mealybugs carried *P. citrinum* after 3 d (Fig. 3i). In contrast, none of the mealybugs released onto *P. citrinum*-free honeydew carried *P. citrinum* (Fig. 3i). Furthermore, when we sampled mealybugs without pellets from tomatoes in the field and removed their legs, no *P. citrinum* could be isolated (Supplementary Fig. 5a, c, e). Conversely, mealybugs carrying pellets under the same treatment were found to carry *P. citrinum* (Supplementary Fig. 5b, d, f). Therefore, the *P. citrinum* in pellets can be transferred to mealybugs' honeydew, colonize and proliferate there, and the *P. citrinum* in honeydew can also be transmitted as spores to mealybugs.

### *P. citrinum* can compete with *A. lecanii* in honeydew

Considering that *P. citrinum* proliferating in mealybug honeydew might meet other microorganisms and thus interact with each other, we investigated the competition between *P. citrinum* and *A. lecanii*, a pathogen of mealybugs, which is also capable of colonizing honeydew[29]. As compared with either fungal species inoculated alone, the colony-forming units (CFU) of *A. lecanii* significantly decreased in co-inoculation ($P < 0.001$), while *P. citrinum* proliferated at a significantly greater rate in co-inoculation ($P < 0.001$; Fig. 3j). These findings suggest that *P. citrinum* is well-suited for thriving in honeydew, potentially outcompeting *A. lecanii*, likely due to previously demonstrated antibiosis (Fig. 2e, k) and possibly a faster growth rate of *P. citrinum* in honeydew.

### Compounds in pellets can inhibit certain pathogens

Because pellets are present with mealybug waxy filaments (Fig. 1b, c) and secretions of tomato glandular trichomes (Fig. 1g), and pellets also possibly contact the microhabitats harboring *P. citrinum* (e.g., honeydew), pellets might have certain compounds capable of inhibiting mealybug or plant pathogens, such as 2,4-Di-Tert-Butylphenol (2,4-DTBP), which can be synthesized by both plants and fungi and possess broad-spectrum antimicrobial properties[32–34]. To test this hypothesis, we investigated the presence of 2,4-DTBP in pellets, where this compound probably exists as a component of the encompassed waxy filaments[35]. We also investigated the presence of the 2,4-DTBP in tomato leaves and PCF products, which might

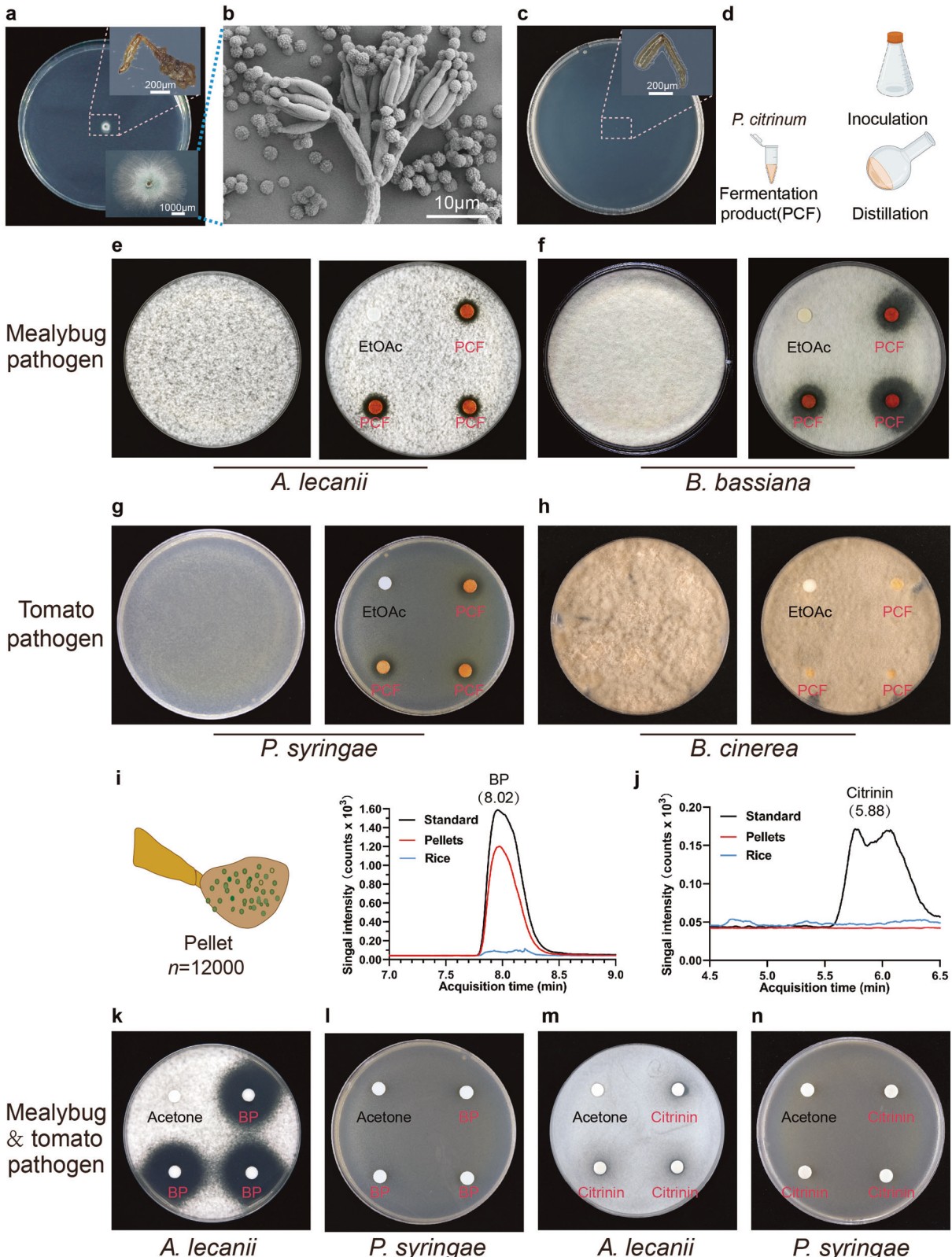

**Fig. 2 | Morphology and antimicrobial properties of *Penicillium citrinum* identified from pellets. a** *P. citrinum* colony generated from a mealybug leg with pellet on PDA plate. **b** Morphology of *P. citrinum* under SEM (scanning electron microscopy). **c** No fungi were isolated from legs without pellets on PDA plate. **d–h** Antimicrobial ability assays of *P. citrinum* fermentation products (**d**) against to two mealybug pathogens, *Akantaomyces lecanii* (**e**), *Beauveria bassiana* (**f**), and two tomato pathogens, *Pseudomonas syringae* (**g**) and *Botrytis cinerea* (**h**), using the disk-diffusion method; the left microbe-containing plate is a blank control. **i**, **j** Butylparaben (**i**) and citrinin (**j**) identified in pellets and rice extract with LC-MS analysis. **k**, **l** Inhibition caused by butylparaben against *A. lecanii* (**k**) and *P. syringae* (**l**). **m**, **n** Inhibition caused by citrinin against *A. lecanii* (**m**) and *P. syringae* (**n**). EtOAc ethyl acetate, PCF *P. citrinum* fermentation product, BP butylparaben.

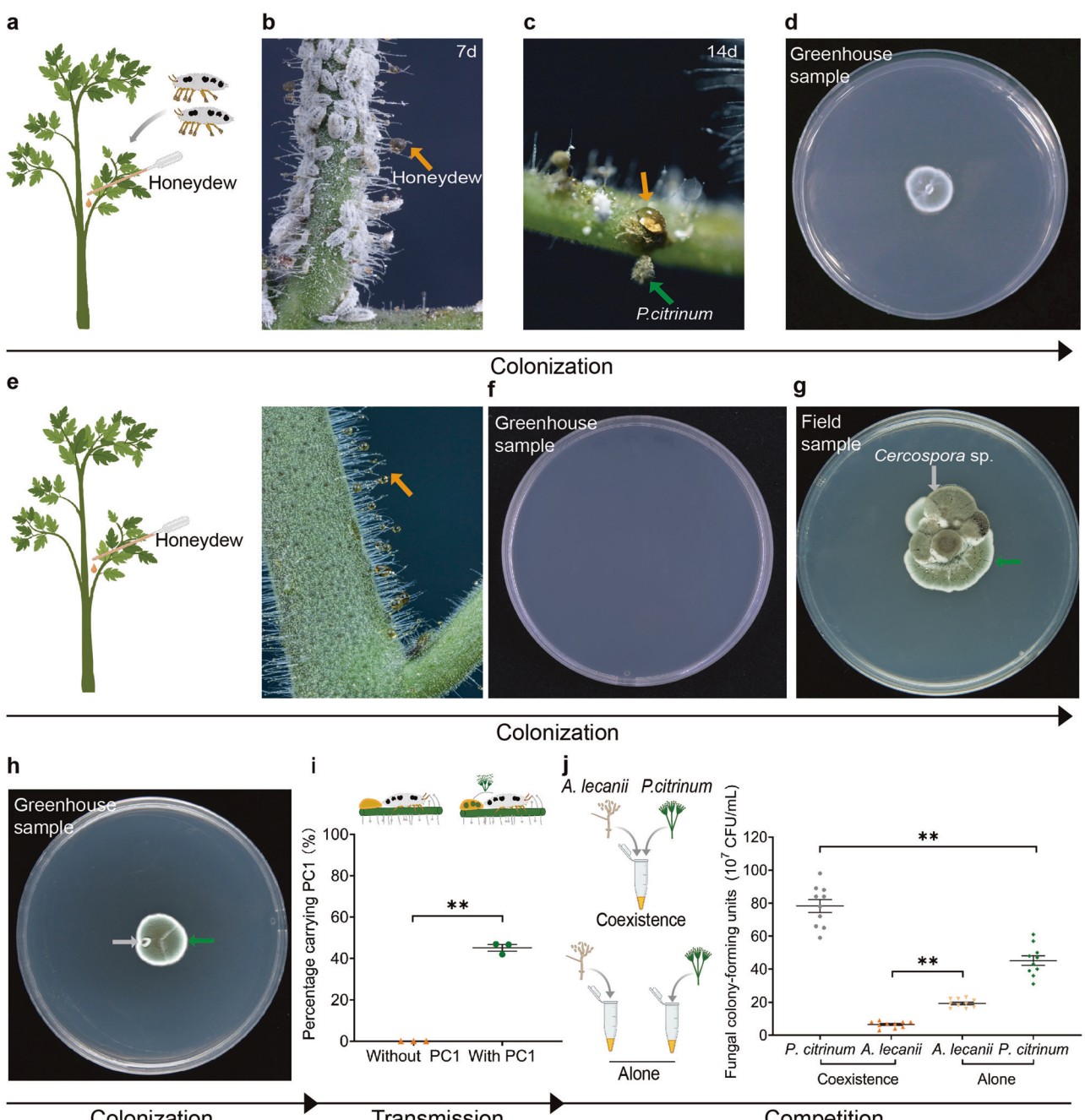

**Fig. 3 | Colonization, transmission, and competitiveness of *P. citrinum* in honeydew. a–c** Proliferation of *P. citrinum* (indicated with green arrows) in mealybug honeydew (indicated with orange arrows) on the tomato plants previously receiving sterile honeydew and pellets-carrying mealybugs treatment (**a**), at 7 d (**b**) and 14 d (**c**) after treatment, and the *P. citrinum* generated from the honeydew on PDA plate (**d**). **e, f** No fungi were found in the honeydew applied to mealybug-free tomatoes at 7 d (**e**), and no fungi were isolated from honeydew cultured on PDA (**f**). **g, h** *P. citrinum* and *Cercospora* sp. (indicated with green arrow and gray arrow, respectively) isolated from honeydew sampled from the tomato plants previously treated with honeydew and mealybugs in the field (**g**), and from the honeydew sampled from tomato plants infested with mealybugs in the greenhouse (**h**). **i** Percentage of mealybugs carrying *P. citrinum* (PC1) after 3 d of feeding on the tomato previously treated with microbe-free honeydew or with *P. citrinum*-inoculated honeydew (30-32 insects per replicate). **j** Competition between *P. citrinum* and *A. lecanii* after 48 h of incubation in the honeydew. Data were presented as the mean (horizontal line) ± SE while points represent each biological replicate. The number of asterisk represent the *p* value (*$P < 0.05$; **$P < 0.01$) as determined by Student's *t* test (**i, j**).

be included in pellets through contact. Then, we assayed in vitro the antimicrobial ability of 2,4-DTBP against two pathogens of mealybug, *A. lecanii* and *B. bassiana*, and two pathogens of tomato, *P. syringae* and *B. cinerea*. In addition, we investigated the presence of BP and citrinin in pellets, the two dominant components of PCF products (Supplementary Table 1), which have been proved of antimicrobial ability against *A. lecanii* (Fig. 2k), or against other microbes[28,36].

As anticipated, 2,4-DTBP was detected in pellets at a concentration of 0.087 ± 0.002 mg per gram of pellets (Fig. 4a and Supplementary Fig. 6a), as well as in tomato leaves and PCF products (Fig. 4b, c). Solutions containing 2,4-DTBP at concentrations of 0.1 and 0.08 mg/mL exhibited strong inhibition against *A. lecanii* and *B. bassiana*, whereas concentrations of 0.01 and 0.001 mg/mL did not (Fig. 4d, e and Supplementary Fig. 6b, c). No inhibitory activity was observed for either concentration of 2,4-DTBP against *P.*

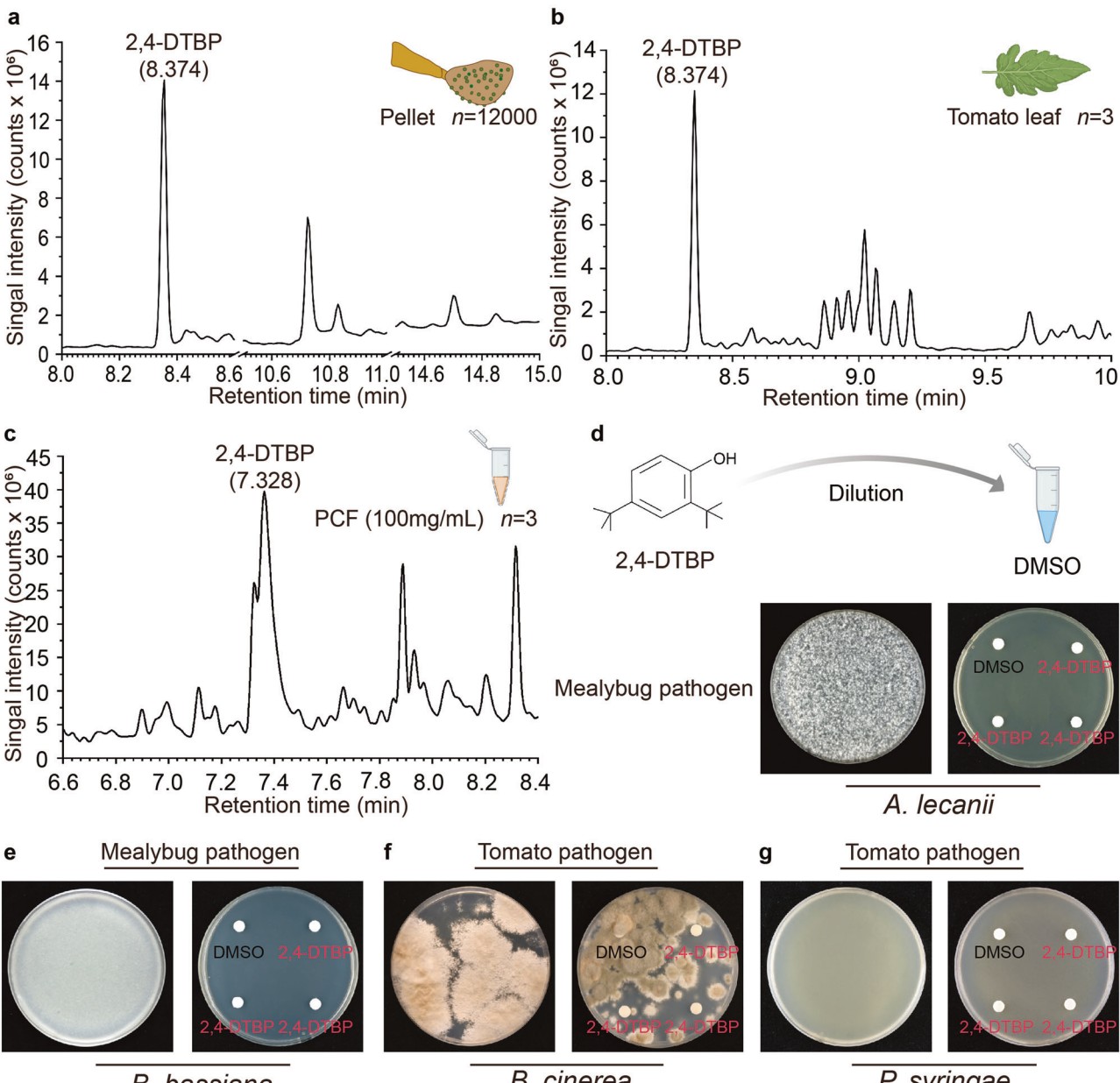

**Fig. 4 | Distribution and antimicrobial activity of 2,4-Di-Tert-Butylphenol.** 2,4-DTBP identified in pellets (**a**), tomato leaves (**b**), and fermentation product of *P. citrinum* (**c**) with GC-MS analysis. **d–g** Inhibition activity of 2,4-DTBP against *A.* *lecanii* (**d**), *B. bassiana* (**e**), *B. cinerea* (**f**), and *P. syringae* (**g**). Plates containing only corresponding microbes as a control (left) vs. plates with both microbes and antimicrobial compounds (right). Control paper disks contained only DMSO.

*syringae* and *B. cinerea* (Fig. 4f, g). BP was also detected in pellets (Fig. 2i), but citrinin was not (Fig. 2j). These findings suggest that the presence of 2,4-DTBP and BP in pellets confers certain antimicrobial properties against mealybug pathogens.

**Pellets help to increase the survival rate of mealybugs under pathogen pressure**

In order to clarify the protective roles of pellets for mealybugs, we compared the survival rate of mealybugs carrying pellets with those without pellets on the tomato leaves supplied with *A. lecanii* spores. After 7 d after treatment, the mealybugs carrying pellets survived at 95.7 ± 1.5%, which was significantly higher ($P < 0.001$) than those without pellets (74.3 ± 1.4%; Fig. 5a and Supplementary Fig. 7). This indicates that the presence of pellets at leg can effectively protect mealybugs from pathogen attack.

**Pellets matter with the antiherbivore defense response of host plants**

As pellets are located at the tarsus and pretarsus of mealybugs and thus contact plant surface, we suspected that they might be involved in interactions between mealybugs and plants. The crawling activity of *P. solenopsis* resulted in physical damage to glandular trichomes on tomato leaves, regardless of the presence of pellets (Supplementary Fig. 8), which, as suggested by Steffens and Walters[15] and McDowell et al.[37], would likely trigger the secretion release from trichomes. However, crawling by mealybugs (without feeding or defecation during crawling) with pellets led to a significantly lower expression level of proteinase inhibitor 2 (*PIN2*, a defense gene in tomato ready to be induced for expression upon insect crawling[38]) in tomato leaves compared to crawling by mealybugs without pellets ($F_{(1,4)} = 152.2$, $P < 0.001$) at 12 h mealybugs were released onto the leaves. The *PIN2* expression level was also significantly lower at 24 h ($F_{(1,4)} = 48.6$,

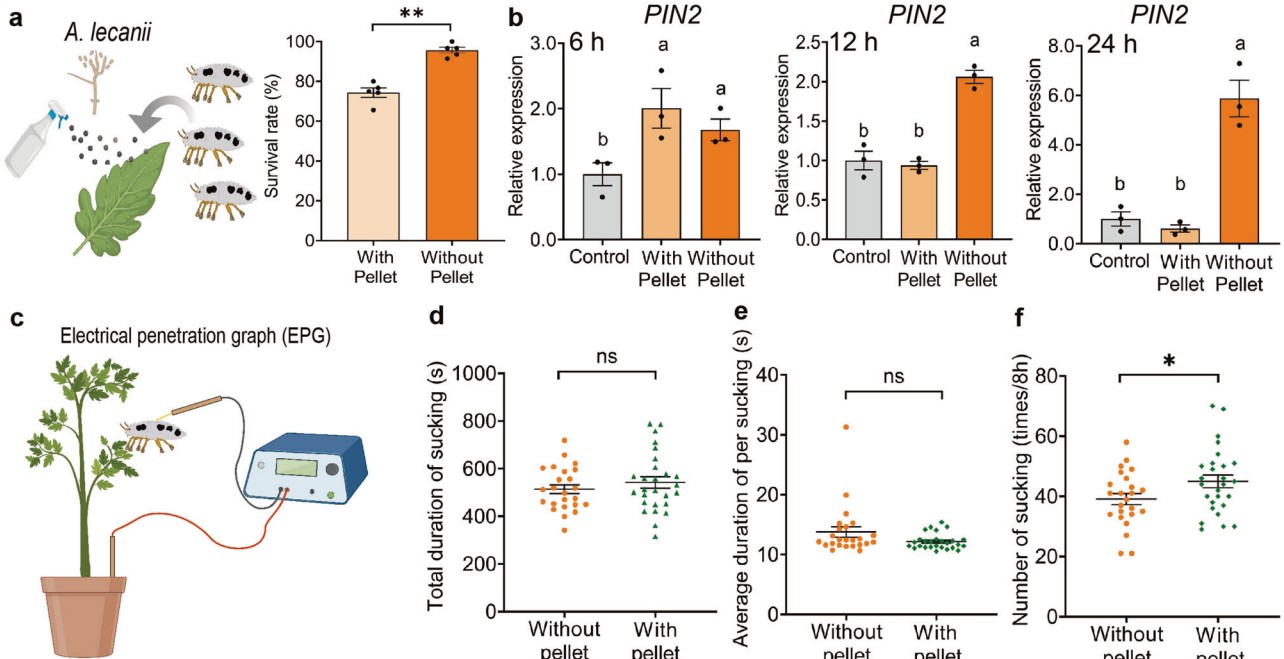

**Fig. 5 | Biological functions of the pellets at *P. solenopsis* legs. a** Survival rate of mealybugs with pellets (29–32 insects per replicate) and without pellets (18–26 insects per replicate) after 7 d of exposure to leaves sprayed with *A. lecanii* spores. The number of asterisk represent the *p* value (**P* < 0.05; ***P* < 0.01) as determined by Student's *t*-test. **b** *PIN2* gene expression in tomatoes at 6 h, 12 h, and 24 h after 10 min of crawling by mealybugs with pellets (3 groups of tomato leaf powder per replicate, with each group consisting of 5 leaves from different tomato plants) and by those without pellets (3 groups per replicate). The relative expression level was normalized using the housekeeping gene *TIP41* as the baseline and visualized as mean ± SE. Treatments with the same letters are not significantly different from each other at alpha = 0.05 based on ANOVA followed by a post-hoc Tukey's multiple comparison test. **c–e** EPG system (**c**) was used to record total duration (d), average duration (**e**), and frequency (**f**) of sucking by mealybugs with (*n* = 27 insects) and without (*n* = 24 insects) pellets over 8 h. The asterisk represents *P* < 0.05 as determined by Student's *t* test and Mann–Whitney *U* test.

*P* = 0.002; Fig. 5b). Hence, the presence of pellets on the legs can potentially reduce the anti-herbivore defense response of tomato to mealybugs.

### The presence of pellets increased the sucking frequency of mealybugs

We also investigated whether the presence of pellets could affect mealybugs' sucking performances, by comparing mealybugs carrying pellets with those without pellets using the electrical penetration graph (EPG) technique (Fig. 5c). The two groups of mealybugs displayed similar waveforms (Supplementary Fig. 9) and did not differ significantly in the total sucking duration and mean sucking duration (Fig. 5d, e). However, the pellet-carrying group had a significantly higher mean sucking frequency than the group without pellets (Fig. 5f).

### Discussion

The knowledge of where insect-associated fungi reside is limited to a few insect taxa[1]. Here, we report a type of pellet at the leg of mealybugs (at-leg pellets) that harbor *P. citrinum*, a fungus capable of suppressing pathogens of mealybugs and plants. Particularly, the pellets are formed by mealybugs in such a kind of plants (*Solanum*) that typically use the secretions from glandular trichomes to defend herbivory insects[39], as well as various microorganisms[17,40]. Besides, such pellets appear to be involved in mealybugs' feeding behavior and their hosts' defense. These findings are very interesting in the fields of insect ecology and interspecific interactions.

The formation of pellets likely varies depending on the host species. Our study revealed that *P. solenopsis* did not develop pellets on their legs when feeding on potato and eggplant (Supplementary Fig. 1), which have fewer types of glandular trichomes compared to tomato[19,22,23]. Different types of glandular trichomes are known to produce varying kinds and amounts of secretions[19]. Additionally, within a plant species, differences in plant types (e.g., wild vs. cultivated) may also influence pellet formation due to variations in glandular trichome density on their surface[22]. Further investigation into cases of mealybugs carrying pellets on their secretion-producing hosts (including wild and cultivated plants) is warranted, such as examining the situation of *P. solani* on tobacco, which possesses glandular trichomes[41,42], and of *P. marginatus* on eggplant, which also harbors trichomes[23,43].

The most interesting finding of our study is the protective features of pellets conferred to mealybugs. First, pellets likely prevent pathogens from infecting mealybugs (e.g., *A. lecanii*; Fig. 5a), as the tarsi and pretarsi where the pellets are located (Fig. 1a), are the primary infection sites of pathogens[44]. Secondly, the pellets, containing spores of *P. citrinum*, serve as a reservoir of this fungus, potentially benefitting mealybugs when the fungus is released from pellets into honeydew, where it proliferates and suppresses mealybug pathogens like *A. lecanii*. *P. citrinum* may possess broad antimicrobial properties, producing substances like BP, which exhibited antimicrobial ability in our study (Fig. 2k), and citrinin, known for its diverse bioactivities, including antimicrobial effects[36]. However, our study did not confirm the antimicrobial activity of citrinin against *A. lecanii* and *P. syringae*. Citrinin has been described as a defense metabolite of *Penicillium corylophilum* when stressed with the antagonist fungus *B. bassiana*[45], some strains of which are mealybug pathogens[46,47]. This suggests citrinin may have the potential to protect mealybugs from pathogens, though our study did not demonstrate this, and the reasons remain unclear. Additionally, pellets likely serve as the primary source of *P. citrinum* in mealybugs' microhabitats; as mealybugs disperse among or within plants, *P. citrinum* disperses along and has the opportunity to proliferate. In contrast, plants and pellet-free mealybugs may be negligible as sources of *P. citrinum*, as we were unable to isolate this fungus from them. However, this does not necessarily mean they do not harbor *P. citrinum*; the fungus may be present in very small quantities. Thirdly, pellets themselves seem unsuitable for harboring pathogens, because some of their components inhibit certain microorganisms,

including BP and 2,4-DTBP. Notably, citrinin was not detected in pellets (Fig. 2j). There are three possible reasons for this: (a) citrinin may not be synthesized by *P. citrinum* at the spore stage, according to Touhami et al.[48], thus it may be absent in pellets despite the presence of spores; (b) during pellet formation, citrinin may be present elsewhere (e.g., honeydew colonized by *P. citrinum*), but not incorporated into pellets; and c) during pellet formation, *P. citrinum* may not be at the stage of producing citrinin.

Honeydew attracted much attention in our study, because it is secreted successively by mealybugs, often abundant, and reported to be involved in interactions of aphid[49], mealybug[50] with their symbiotic fungi. Previous studies have indicated that the honeydew of mealybugs might provide substrates for the growth of their symbiotic fungi[50,51]. Our findings align with this (Fig. 3a–d), demonstrating that *P. citrinum* present in honeydew can be transmitted to mealybugs, and honeydew acts as a milieu for *P. citrinum* to interact with microorganisms, including pathogens of mealybugs and plants. This enriches our knowledge of the functions of honeydew, and provides an example of how the materials produced by insects (feces, honeydew, etc.) could be used by fungi to promote their symbiosis with partners.

In honeydew, *P. citrinum* exhibits better growth in a mixture with *A. lecanii* compared to when it grows alone, while *A. lecanii* shows reduced growth (Fig. 3j), indicating potential benefits for *P. citrinum* from *A. lecanii*. This phenomenon could be attributed to two possible reasons. Firstly, antibiosis might play a role, where *P. citrinum* secretes certain compounds, such as butylparaben, potentially inhibiting the growth of *A. lecanii* to some extent (Fig. 2k). Secondly, the shift in *P. citrinum*'s reproductive strategy could occur in response to encountering a more limited resource. According to Chan et al.[52] and Gilchrist et al.[53], fungi tend to allocate more energy to reproduction as available resources deplete. Thus, when coexisting with *A. lecanii* in honeydew, *P. citrinum* might reproduce more rapidly than existing aloneas a response to a rapider consumption of available resource.

The EPG data suggests that the presence of pellets on the legs may result in an increase in sucking frequency (Fig. 5f), although the underlying mechanisms remain unclear. However, considering that secretions from the tarsi of insects can trigger plant defense mechanisms[54], one possible explanation could be that as the tarsi and pretarsi are laden with pellets, mealybugs would have less direct contact with glandular trichomes, thereby reducing the induction of plant defense genes. This reduction in defense gene expression may favor an increase in sucking frequency[55,56]. This hypothesis finds support in the observed decrease in *PIN2* expression levels in plants subjected to crawling by pellet-carrying mealybugs (Fig. 5b). Additionally, other studies, such as Alvarez et al.[57], have found that aphids (*Myzus persicae*) feed more efficiently (with a higher sucking frequency) on potato leaves where glandular trichomes have been removed compared to leaves with intact trichomes. Thus, pellets play a role in the interactions between mealybugs and their hosts, not only providing protection from pathogens but also affecting mealybug behavior. In future research, exploring additional metrics related to mealybug behavior, such as the accumulation of defensive compounds in leaves following exposure to mealybugs with or without pellets, could provide further insights.

Only one culturable fungus, *P. citrinum*, was consistently isolated from pellets and honeydew (Figs. 2 and 3), despite different types of culture medium being tried. Yet, it is not surprising, because *P. citrinum* can inhibit fungi and some other factors also have antimicrobial property in in vitro assays, such as 2,4-DTBP in wax of mealybugs[12] (Fig. 4), and the secretions of plants[58]. Such a *P. citrinum*-dominating fungal community may have contributed a great deal to mealybugs' success, also somehow to hosts, by inhibiting their pathogens.

In summary, we found a kind of pellets at legs of insects that associated with *P. citrinum* and provide multiple protections to mealybugs, through the combination of antimicrobial compounds from the fungus, insect wax, and host secretions, or the involvement during insect dealing with host defenses. To the best of our knowledge, this is a novel finding in the contexts of insect-microbe-plant tripartite associations, and would especially deepen our understanding of how insects use associated fungi to adapt to environments, including host and pathogen stresses.

## Materials and methods

### Insects, plants, and microorganisms

Cotton mealybugs (*P. solenopsis*) were collected from *Hibiscus mutabilis* plants in Hengfan, Lanxi, Zhejiang Province, China; Solanum mealybugs (*P. solani*) were collected from *Lithops* sp. in Yushi Valley, Hangzhou, Zhejiang Province, China; and papaya mealybugs (*P. marginatus*) were collected from papaya *Carica papaya* adjacent to Hainan University, Danzhou, Hainan Province, China. The collected insects were identified based on their morphology and molecular biology[12,41,43]. The collected insects were reared in screened cages (40 cm × 50 cm × 50 cm) at the Zijingang Campus of Zhejiang University (ZU) with potted tomato and cotton for 2–4 generations prior to use in a phytotron maintained at 26 ± 1 °C and RH 70 ± 5% with a photoperiod of 14 h : 10 h (L : D). *P. solenopsis* were also reared in cages with potted tomato in a greenhouse (26 ± 5 °C) and a field at the campus.

Tomato (*S. lycopersicum*, cv. Cooperative 903), cotton (*G. hirsutum*, cv. Zhemian 1793), potato (*S. tuberosum*, cv. Netherlands Fifteen), and eggplant (*S. melongena*, cv. Yinqie) were cultivated in the phytotron set under same conditions as described above. Tomatoes were also grown in a greenhouse (26 ± 5 °C) and a field at the Zijingang Campus of ZU.

Tomato bacterial pathogen, *P. syringae* pv. tomato isolate DC3000, and fungal pathogen *B. cinerea* were supplied by Prof. Fengming Song and Dr. Dayong Li of ZU. All of the other microbial strains used in this study were purchased from the China General Microbiological Culture Collection Center (CGMCC), including the mealybug pathogenic fungal strains *A. lecanii* (CGMCC NO. 3.4505)[44] and *B. bassiana* (CGMCC NO.3.15729)[46]. The fungi and bacteria were incubated on PDA plates and LB plates, respectively, and maintained under 27 ± 1 °C in a laboratory incubator.

### Characterization and source of pellets on mealybug legs

Stereo microscopy (Nikon, Tokyo, Japan) was used to observe the location of the pellets on female adult legs of *P. solenopsis*, *P. solani*, and *P. marginatus* that were reared on tomato plants under phytotron, while some *P. solenopsis* female adults from field-grown tomato were also observed. At least 30 individuals were observed for each species. To determine whether the pellets were lost at molting and reattained afterwards, 30 newly molted *P. solenopsis* female adults (continuously reared on tomato leaves) were observed at intervals of 0.5 h under a stereo microscope for the presence of pellets. The surface and internal structure of pellets sampled from *P. solenopsis* female adults were observed using scanning electron microscopy (SEM). Briefly, legs with pellets were taped onto a stub and dried in an ion sputter (Hatachi, Tokyo, Japan) under vacuum, and after gold sputtering, they were observed under a TM-1000 SEM (Hatachi).

The probability of attaining pellets through generations was observed by transferring reproducing *P. solenopsis* females which were previously fed on cotton (at the four-leaf stage, where no pellets can be formed; the same below) to tomato plants (at the four-leaf stage), and rearing them for two successive generations in the greenhouse described above. At each generation, the legs of adult females were observed and the number of pellets was recorded for each individual. The observation was replicated five times for each generation, with at least 20 mealybugs observed in each replicate.

In the experiment comparing the percentage of *P. solenopsis* female adults attaining pellets on intact tomato leaves and the leaves with their secretions removed previously, we sampled two groups of tomato leaves (at the five-leaf stage) and prepared as follows. One group was left intact. The other group was placed in ultrapure water and cleaned in an ultrasonic cleaner (Saentz, Ningbo, China) at 30 Hz at 25 °C for 10 min[59] to remove secretions on the leaf surface, and then the leaves were placed in plastic boxes and maintained at room temperature until the water on the leaf surface fully evaporated. Next, leaf petioles of both groups were wrapped with moist cotton, released with *P. solenopsis* female adults (lacking pellets) previously

fed on cotton, and maintained in the phytotron described above. After 12 h, all leaves in each group were replaced with fresh leaves from the corresponding treatment. After 24 h, the formation status of pellets at leg was observed under a stereo microscope for each mealybug. The experiment was replicated five times.

Observation of pellet formation on cotton, potato and eggplant were performed by releasing *P. solenopsis* 1st-instar nymphs (lacking pellets) previously fed on cotton to each of these plants (at the four-leaf stage), and then maintaining the plants in the phytotron described above. After the mealybugs grew to 3rd instar, mealybugs were sampled at five different time for observation of pellets. At each time, at least 30 mealybugs were used from each plant.

### Identification of microbes in at-leg pellets

The legs with pellet were sampled from *P. solenopsis* adult females living on tomato under field and phytotron conditions, and inoculated on PDA plates. As a control, legs without pellets were also inoculated on PDA plates, which were sampled from the mealybugs feeding on secretions-free tomato leaves prepared using procedures described above. Both legs with pellets and those without pellets from field-sampled mealybugs were inoculated on CzA, YPD, and SDA plates. Incubation of all plates occurred at $27 \pm 1$ °C, and fungal isolates were examined after 72 h. The morphology of isolated fungi was observed using SEM.

DNA was extracted from isolated fungi using the HP Fungal DNA Kit (OMEGA Bio-Tek, Norcross, USA) reagent kit, according to the manufacturer's instructions. Extracted DNA was used for PCR-based molecular identification of the fungi. PCR was performed using an Applied Biosystems 2720 thermal cycler PCR (Thermo Fisher Scientific, Waltham, USA), and a 50 µL reaction containing 2 µL DNA, 21 µL $H_2O$, 1 µL each of two primers, and 25 µL 2×GoTaq Green Master Mix (Promega, Madison, USA). The primers target a 550 bp ITS of fungi: ITS-1 5′-CTTGGTCATTTA-GAGGAAGTAA-3′ (specific for higher fungi) and ITS-4 5′-TCCTCCGCTTATTGATATGC-3′ (universal primer)[60–62]. PCR cycles were as follows: pre-denaturation at 94 °C for 4 min; denaturation at 94 °C for 40 s, annealing at 55 °C for 50 s, and extension at 72 °C for 1 min, for 35 cycles; and a final extension at 72 °C for 10 min. As there may be more than one ITS copy in fungal genomes[63] and thus the ITS could possibly not be identified accurately through direct sequencing, the PCR product was cloned using pGEM®-T Easy Vector Systems (Promega) and sequenced by Tsingke Biotech (Hangzhou, China). To determine the fungal species, the ITS sequences were blasted in NCBI; reference sequences retrieved from GenBank (Supplementary Table 2) were used for phylogenetic analysis (Maximum likelihood method with 1000 bootstrap replications) using MEGA 7[64].

### Antimicrobial activity assays of *Penicillium* discovered in at-leg pellets

The *P. citrinum* fermentation (PCF) products used in assays were extracted following the procedures published by Wang et al.[27]. Briefly, PDA liquid medium containing *P. citrinum* was inoculated into sterilized rice, incubated for 14 d, and then extracted with ethyl acetate (EtOAc). Then, the disk-diffusion method was used to test the antimicrobial activity of PCF products against two fungal pathogens of mealybugs, i.e., *A. lecanii* and *B. bassiana*, a tomato bacterial pathogen *P. syringae*, and a tomato fungal pathogen *B. cinerea*. The PCF disks (Blank Cartridges; Thermo Fisher Scientific, Waltham, USA) were saturated with PCF products at 100 mg/mL, and EtOAc disks were saturated with EtOAc as control. Spores of *A. lecanii*, *B. bassiana* and *B. cinerea* were added separately to PDA plates to reach a concentration of $10^4$ spores/mL each. Then, three PCF disks and one EtOAc disk were placed evenly on each of these plates. For the assay with *P. syringae*, LB plates were used, with each containing $10^7$ CFU/mL of *P. syringae*. Plates containing only a pathogenic isolate and no-disk plates were used as control. To ascertain whether the identified antimicrobial activity stemmed exclusively from *P. citrinum* rather than from the rice, we subjected rice to the same extraction procedure used for preparing PCF products, with the exception of omitting *P. citrinum*. The resulting rice extract (RE + PDA liquid media)

was then tested for antimicrobial activity against the four pathogens using the disk-diffusion method, with EtOAc disks serving as a control. All antimicrobial assays adhered to the standards established by the Clinical and Laboratory Standards Institute (CLSI, 2010; 2021)[65,66].

The primary components of PCF products were examined by LC-MS/MS using a Vanquish UHPLC-MS/MS system (Thermo Fisher Scientific). The raw data files generated by this system were processed using Compound Discoverer 3. 1 (CD3.1, Thermo Fisher Scientific) to perform peak alignment, peak picking, and quantitation for each metabolite. Then, peaks were matched with the mzCloud[67], mzVault[68], and Masslist[69] to obtain accurate qualitative and relative quantitative results. After matching the peaks of each compound in the PCF products, we identified the compounds with the highest content, i.e., BP and citrinin. The presence of BP and citrinin in RE was also examined by LC-MS/MS using the Infinity LC Clinical Edition/K6460 (Agilent, Palo Alto, USA), through the comparisons of retention time, molecular ion peaks, and specific fragment ions between the standard compounds and the compounds in RE. Antimicrobial activity of BP and citrinin (Sigma, St. Louis, USA) were tested using the disk-diffusion method described above, except that acetone was used as the solvent to dissolve the two compounds to a concentration of 100 mg/mL.

Plates containing bacteria were incubated at $27 \pm 1$ °C for 48 h, and those containing fungi were incubated at $27 \pm 1$ °C for 72 h. Each assay was conducted three times. The inhibition circles were measured to determine the antimicrobial activity.

### Colonization and proliferation of *P. citrinum* in honeydew and horizontal transmission to mealybugs

Honeydew of *P. citrinum* was collected from tomato plants maintained in the phytotron using a sterilized 10 µL pipette tip and mixed with distilled water at a ratio of honeydew : water = 1 : 2, v/v. After sterilization by sterile syringe filter, 50 µL of diluted honeydew was evenly smeared to the same stem position on non-infested tomato plants using six replicate plants. The plants were then divided into two groups and placed in two cages (40 cm x 50 cm × 50 cm). One group was left free of insects as the control, while the other group was each released with 180 pellets-harboring 2nd-instar nymphs of *P. solenopsis*. Thereafter, the two cages were maintained in the greenhouse, and the growth of fungus in the honeydew was observed daily. When fungal colonies became visible in honeydew (at approximately 7 d), the honeydew from plants of each group was sampled (at approximately 14 d), transferred to PDA plates, and incubated at $27 \pm 1$ °C for 72 h. In order to determine whether *P. citrinum* could colonize *P. solenopsis* honeydew under natural conditions, honeydew was also collected from *P. solenopsis*-infested tomato plants in the field and greenhouse, and used to inoculate PDA plates as described above. All fungi from honeydew were observed under a microscope for colony and hypha morphology, and their ITS sequences were cloned, to determine their species.

To observe *P. citrinum* transmission from honeydew to *P. solenopsis*, honeydew inoculated with $10^7$ spores/mL of *P. citrinum*, and honeydew not inoculated (control), were prepared and cultured on a shaker at $27 \pm 1$ °C and 200 r.p.m for 24 h. Then, 50 µL of each kind of honeydew was applied separately to tomato leaves, and each leaf was infested with 50 3rd-instar *P. solenopsis* nymphs without pellets (previously reared on cotton). Before use, all of the tomato leaves and nymphs were free of *P. citrinum*, as determined by inoculating their homogenates on the PDA plates. After infestation, the leaves were placed in autoclaved plastic boxes (separate boxes for the inoculated and control honeydew) and kept at $27 \pm 1$ °C on a clean bench to minimize chances for contamination. After 3 d, 30 mealybugs were randomly sampled from boxes, ground, and 50 µL of the grinding solution was applied to PDA plates and incubated at $27 \pm 1$ °C for 72 h. The number of *P. citrinum* colonies on PDA medium was then counted, and the percentage of mealybugs carrying *P. citrinum* was calculated. All experiments were performed in triplicate.

To determine whether the *P. citrinum* in the environments (e.g., honeydew) could be transmitted to other parts of mealybugs besides their legs, 60 *P. solenopsis* 3rd-instar nymphs (roughly half of them carrying pellets

at leg) were randomly sampled from tomato in the field. After being removed with legs, they were ground, and 50 μL of the grinding solution was applied to CzA, YPD, and SDA plates. The plates were incubated at $27 \pm 1$ °C for 72 h. All fungal isolates were identified based on ITS sequences.

### Competitive ability of *P. citrinum* with *A. lecanii* in honeydew
To create the mixed suspensions required for this assay, spore suspensions of *P. citrinum* and *A. lecanii* at $10^8$ spores/mL were mixed and diluted with sterile water until each fungus reached a concentration of $10^7$ spores/mL. Single isolate suspensions of $10^7$ spores/mL were prepared in a similar way. Then, 25 μL of each of the three suspensions was added separately to 25 μL of diluted honeydew, and the inoculated honeydew was incubated on a shaker at $27 \pm 1$ °C and 170 r.p.m for 48 h. Next, the incubated honeydew cultures were diluted 1000-fold, and 50 μL was spread on PDA plates, which were then incubated at $27 \pm 1$ °C for 72 h. After this second incubation, the number of *P. citrinum* and *A. lecanii* colonies was counted, and their CFU in the incubated honeydew cultures was calculated. The experiment was conducted ten times. The strain of *B. bassiana* (some of its strains are pathogens of mealybugs) selected for our study was excluded from this assay (as well as subsequent experiments) because it did not induce mortality in mealybugs during our preliminary testing.

### Compounds in pellets and their antimicrobial activity
To determine the presence of 2,4-DTBP in pellets, 12,000 pellets were collected from *P. solenopsis* female adults and placed in 1.5 mL glass tubes; a total of three such samples were prepared. Hexane (99.9% purity) was added to each sample to reach a pellet concentration of 100 mg/mL. Then, the samples were vortexed and left at room temperature for 30 min, followed by ultrasonic vibration for 30 min to ensure the pellets were fully dissolved. The dissolved pellet solutions were centrifuged at $15,000 \times g$ for 2 min, and the supernatants were transferred to new 2-mL sealed amber glass vials.

The examination of the presence of 2,4-DTBP in tomato plants and PCF products were performed using previously published methods. Briefly, the process involved freezing and drying the tomato leaves, grinding them into a powder, and dissolving them in hexane[70]. For the PCF products, it was concentrated under vacuum and then directly dissolved in ethyl acetate after centrifugation[71]. Three replicates of both tomato plants and PCF products were used for analyses. Standard quality 2,4-DTBP (Sigma) was spiked with hexane to make a series of standard solutions (50 to 6400 ng/μL). The supernatant of the centrifuged samples of dissolved pellets, tomato leaves and PCF products, and 2,4-DTBP solutions of each concentration was analyzed on a TRACE 1310 (Thermo Fisher Scientific) gas chromatograph (GC) equipped with an ISQ single quadruple MS (Thermo Fisher Scientific) and interfaced with the Chromeleon 7.2 data analysis system (Thermo Fisher Scientific), with a constant flow of helium at 1 mL/min. Next, the antimicrobial activity of 2,4-DTBP against the mealybug and tomato pathogens was assayed using the disk-diffusion method as described above. The disks were saturated separately with 0.08, 0.1, 0.01, and 0.001 mg/mL 2,4-DTBP dissolved in DMSO, using disks saturated with DMSO as control. All experiments were performed in triplicate.

The detection of BP and citrinin in pellets was performed using another sample of 12,000 pellets, adopting the methods utilized for assays of RE described above.

### Protection of mealybugs conferred by at-leg pellets
Spore suspensions of *A. lecanii* at $10^7$ spores/mL were evenly sprayed onto both sides of tomato leaves (at the four-leaf stage), with 1 mL spore suspension per leaf. After the evaporation of water from the leaf surface, the leaves were divided into two groups. One group was infested with 30 *P. solenopsis* 3rd-instar nymphs carrying pellets, while the other group was infested with nymphs previously fed on secretions-removed tomato leaves and thus lacked pellets. All leaves were placed in autoclaved plastic boxes and kept in a phytotron maintained at $26 \pm 1$ °C and RH $70 \pm 5$% for 7 d, during which dead mealybugs were collected to confirm whether their death resulted from pathogenic infection. The corpses were first surface sanitized

using 0.6% sodium hypochlorite and 75% ethanol, which has been testified rather effective[72] and frequently adopted by other researches to determine whether the insects die of pathogens or not[73]. Then, the corpses were placed on PDA plates and incubated at $27 \pm 1$ °C for 72 h. The mealybugs were recorded as dead due to *A. lecanii* infection only when the single fungus *A. lecanii* colonized the PDA, as could be identified according to their morphology. The experiment was conducted five times.

### Effects of pellets on the plant antiherbivore defensive response
The damage of *P. solenopsis* to the surface of tomato leaves was observed after allowing mealybugs with pellets and those without pellets to crawl on leaves for 10 min. The observation was performed using an SEM as described above.

To observe the pellets-associated defense response in hosts, tomato plants at the four-leaf stage were divided into three treatment groups, each containing five plants. One group was infested with 3rd-instar mealybug nymphs carrying at-leg pellets on each leg, with each plant receiving 10 such mealybugs on one leaf. Mealybugs were allowed to crawl on the leaves for 10 min. Another group of plants was used as a positive control and were infested with nymphs without at-leg pellets in the same position. To exclude the interference of other factors, nymphs were closely observed, individuals showing signs of sucking or defecating were discarded, and the plant was excluded from analysis. A third group of plants received a treatment of no mealybugs. Plants were maintained in the phytotron as previously described. After each of three time points, 6 h, 12 h, and 24 h, one leaf (crawled over by insects) was sampled from each plant, and the five leaves from each group were bulked together (as one replicate) and immediately frozen in liquid nitrogen.

Then, the sampled leaves were ground into powder in liquid nitrogen, and total RNA was extracted with the SteadyPure Plant RNA Extraction Kit (Accurate Biology, Hunan, China), according to the manufacturer's instructions. After diluting the concentration of total RNA to 800 ng/μl, cDNA was synthesized from total RNA using Eastep® RT Master Mix Kit (Promega), with a 10 μL reaction containing 1 μL of 10-fold diluted total RNA, 7 μL of nuclease-free water and 2 μL 5 × Eastep® RT Master Mix. The reverse transcription was performed by Applied Biosystems™ 2720 thermal cycler PCR (Thermo Fisher Scientific) using a protocol of 15 min at 37 °C followed by 5 min at 98 °C and maintained at 4 °C. Next, using previously published methods[38], the expression level of *PIN2*, which is a highly inducible gene encoding the protease inhibitor 2, a representative defense protein against herbivorous arthropods[74,75], was quantified for each sample by quantitative real-time polymerase chain reaction (qRT-PCR) with GoTaq® qPCR and RT-qPCR Systems (Promega). The qRT-PCR was performed on AriaMx real-time PCR system (Agilent, Palo Alto, USA), using two primers specific to this gene: 5′-GGATTTAGCGGACTTCCTTCTG-3′ and 5′-ATGCCAAGGCTTGTACTAGAGAATG-3′. The housekeeping gene TIP41-interacting protein (*TIP41*) was used (5′-ATGGAGTTTTT-GAGTCTTCTGC-3′ and 5′- GCTGCGTTTCTGGCTTAGG-3′) as the baseline to normalize Ct values[76]. Relative quantifications, with untreated plants as the reference group, were calculated using the $2^{-\triangle\triangle C(T)}$ method. All experiments were conducted three times.

### Feeding behavior of mealybugs with at-leg pellets
The EPG technique was used to compare the sucking behavior of mealybug females carrying pellets with those without pellets. Mealybugs (48 h after molting) were collected at a non-feeding state to ensure an intact state of their mouthparts. They were first starved for 24 h, and then the wax on their backs was removed using a soft brush. Next, an insect electrode was connected to the back of insects using copper wire, silver glue, and gold wire (18 μm diameter), and the plant electrode was inserted around the soil of the host plant. The recordings began at 9:00 a.m. and continued for 8 h. As mealybugs attempt to feed on both phloem and mesophyll cells as they probe[77], we counted all the waveforms *pd* (formed when the mouthparts of mealybugs probed mesophyll cells) and waveforms *E2* (formed when the mouthparts probed the phloem) and recorded the total duration, average duration, and frequency of sucking. The experiment was carried out in a

Faraday cage (2 m × 1 m × 1 m) within the phytotron. The mealybugs which fell from plants were discarded and not used in analyses. A total of 30 mealybugs were used to infest each replicate plant.

## Statistics and reproducibility

Data were statistically analyzed using SPSS 25 (SPSS Inc., Williston, USA), GraphPad Prism 9.0 (GraphPad Software Inc., San Diego,USA). Before analyses, data were checked for normal distribution using the Shapiro-Wilk test. Student's $t$-test was performed to compare: percentage of mealybugs forming pellets at F1 (20-91 insects per replicate, 5 replicates) and F2 (48-147 insects per replicate, 5 replicates) generation (Fig. 1e), percentage of mealybugs that formed more than three pellets at the F1 and F2 generation (Fig. 1f), percentage of mealybugs (19-29 and 23-30 insects in per replicate, 5 replicates) forming pellets on intact or ultrasonic surface cleaned tomato leaves (Fig. 1g), percentage of mealybugs carrying $P.$ $citrinum$ after 7 d of feeding on tomato previously treated with microbe-free honeydew or honeydew (30–32 insects per replicate, 3 replicates) inoculated with $P.$ $citrinum$ (Fig. 3i), competition between $P.$ $citrinum$ and $A.$ $lecanii$ after 48 h of incubation in the honeydew (Fig. 3j), survival rate of mealybugs with pellets (29–32 insects per replicate, 5 replicates) and without pellets (18-26 insects per replicate, 5 replicates) after 7 d of exposure to leaves sprayed with $A.$ $lecanii$ spores (Fig. 5a), and total duration and frequency of sucking by mealybugs with ($n$ = 27 insects) and without pellets ($n$ = 24 insects) over 8 h (Fig. 5d, f). Average sucking duration was analyzed using a Mann-Whitney $U$ test (Fig. 5e). Analysis of variance (ANOVA) was used to analyze $PIN2$ gene expression in tomatoes (3 groups per replicate, 3 replicates) receiving different mealybug treatments (Fig. 5b). The graphical illustrations were created with BioRender (BioRender, Toronto, Canada)[78] and Adobe Illustrator (Adobe Systems Inc., San Jose, USA).

## Reporting summary

Further information on research design is available in the Nature Portfolio Reporting Summary linked to this article.

## Data availability

The source data behind the graphs in the paper can be found in Supplementary Data 1. The DNA sequencing data for ITS nuclear rDNA sequences generated from $P.$ $citrinum$ is available under the accession code of GenBank ID: OR647500. And the DNA sequencing data for ITS nuclear rDNA sequences generated from $Aspergillus$ sp. and $Cercosporais$ sp. is available under the accession code of GenBank ID: PP338193 and PP338195. All other data are available from the corresponding author on reasonable request.

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

## Acknowledgements
This work was supported by National Natural Science Foundation of China (32072489; 32102189).

## Author contributions
Zic.L. and M.J. designed the research; Zic.L. performed the research; Zic.L. and H.T. analyzed the data; Zic.L., H.T., and M.J. wrote the paper; M.N., Y.Z., X.Y., Y.T., and Zih.L. collected samples for the insects used to isolate the fungus.

## Competing interests
The authors declare no competing interests.
