## [Peer Review File · Communications Biology]

Reviewers' comments:

Reviewer #1 (Remarks to the Author):

In this manuscript Li and colleagues describe the presence of an at-leg pellet in mealybugs of the species *Phenacoccus solenopsis* growing on tomato plants. This pellet, while originated from host secretions and waxy filaments, also contains a fungal strain: *Penicillium citrinum*. The authors explore the role that these pellets and associated fungi play in (a) defending mealybugs and tomato plants from pathogens, (b) regulating mealybug behavior and (c) host defenses, and (d) colonizing new insect hosts.

This is a fascinating and timely discovery given that fungi are increasingly being recognized as major players in the ecology of insects.

While I enjoyed reading this manuscript, I have major concerns regarding the methodology employed to conclude that the *P. citrinus* strain isolated from mealybugs produces antimicrobial compounds that protect mealybugs against pathogens (see comments below). Likewise, the authors do not mention whether the antimicrobial compounds presumably produced by the fungus are synthesized in the pellet. Therefore, the potential compounds of fungal origin, may not be of ecological relevance.

Other comments:

L37: I am not aware of any example of fungal defense against parasitoids, and I am not sure the cited work features any. The authors may consider verifying this. However, recent relevant studies are missing from the examples such as fungi protecting beetle pupae from predation (Berasategui et al. 2022).

L49: what type of typical insect adaptations are missing in mealybugs?

L63: If Figure 1 is to be a graphical abstract, then Fig. 2a, should just be Fig. 1a...and so on.

L65: It would be helpful to use an arrow to point to the pellet in Figure 2d. Pellets are generally difficult to observe in these photographs.

L91: I find this result very intriguing/interesting. Why would the second generation have more at-leg pellets than the first one? What processes would be driving this phenomenon? Authors could speculate.

L96: I agree with the authors that these results do not demonstrate the negative effect of the pellets on insect performance. This is mainly because the experiment was not set up in a way that could answer that question. Perhaps the authors should consider changing the logic behind this experiment as stated in L86-87.

L104: What is the similarity percentage between the isolated fungal strain to *P. citrinum* from the database?

L107: Did the pellet-free legs failed to produce consistent fungal morphologies, or no fungal isolates at

all?

L160: Given that *P. citrinum* grows better in a mixture with *V. lecanii* than alone, this also suggests that *P. citrinum* is benefiting off *V. lecanii* in some way. Authors could speculate as to why.

L166: Were the compounds isolated from the fermentation of *P. citrinum* (citrinin and butylparaben) also found in these extracts? If not, these compounds may not be produced in natural conditions, and may therefore not be ecologically relevant.

L170: Is Fig2c correct?

Comments regarding methodology:

L280-285: How were the insects reared? (Use of cages, potted plants etc)

L336: What were the growing conditions?

L350: Was the PCR product cloned because there were several ITS copies in the fungal genome?

L362: I have concerns regarding the antimicrobial assays. If I understood it correctly, the authors inoculated the isolated fungal strain onto liquid media, and then this media into sterilized rice. This mixture was then extracted with EtOAc. Treatment disks were generated by dipping disks into this extract mixture. However, control disks were generated by dipping them in EtOAc alone. This suggests that the treatment disks will contain secondary metabolites originating from both the PDA liquid media, the fungal isolate and the rice, whereas the control treatment contains no secondary metabolites. If this is correct, this experiment is unfortunately flawed. The correct control would have been disks dipped in EtOAc extracts originating from PDA liquid media-inoculated rice. Perhaps I misunderstood the methodology, in which case I encourage authors to clarify the procedure.

L371: In the same manner, I have concerns about the experiment exploring the origin of the antimicrobial activity observed. Control extracts were not analyzed, and hence, it is unclear whether the detected compounds butylparaben and citrinin originated from the fungus or rice. In fact, butylparaben has been detected in grains such as barley and flax seed, and is usually employed in industry for its antimicrobial (both antifungal and antibacterial) properties. Citrinin, is indeed produced by some fungi and is known to have antimicrobial properties, but in these experiments, it fails to inhibit pathogens. Additionally, it would be important to explicitly mention that commercial standards were used to perform the antimicrobial assays.

L450: Mealybugs were considered killed by *V. lecanii* if dead animals placed on PDA media grew the pathogen. However, authors do not differentiate between animals that died with *V. lecanii* vs. those that died because *V. lecanii*. An animal with one spore of *V. lecanii* on the surface, that died for a different reason, would still be *V. lecanii* positive on a plate and recorded as killed by the pathogen. Perhaps measuring pathogen abundance via qPCR would give a better estimate.

L209: Correct citation: Biedermann and Vega.

L223: I disagree with the idea that the at-leg pellets explain the mechanism by which mealybugs avoid lethal tomato secretions. Even if the animals redirect plant secretions for the formation of the pellets, these could still be toxic. Authors could consider rephrasing.

L230: I suggest authors to not to overstate the role of at-leg pellets in mealybug overcoming of host plant secretions.

Reviewer #2 (Remarks to the Author):

In this manuscript, Li et al. report the formation of an at-leg pellet derived from tomato secretions and stores the fungus *Penicillium citrinum*. This pellet appears to be an adaptive response of the mealybug to potential trichome toxic secretions and provides a reservoir of the fungus *P. citrinum*, which can protect against fungal diseases in the mealybug. Overall, the paper is well written, the experiments are solid, and the results are clearly presented and discussed. I have some minor comments that could potentially improve the manuscript.

1. The currently accepted name for *Lecanicillium lecanii* is *Akantaomyces lecanii* (Zimmerman) (species fungorum). Correct this throughout the manuscript.
2. Line 47-49 – The two sentences appear to be incomplete. “Causing losses” on what? And “lack many typical adaptations.” these sentences need additional context.
3. It would be good if the authors provided more information about the at-leg pellet in the introduction. This would give a better justification for the study. Indeed, the introduction doesn’t offer any rationale and/or hypotheses. Why is it important to study the function of this “organ.” Are there similar structures formed in other insects? What background information exists about at-leg pellets in mealybugs? Have these at-leg pellets been observed recently? Investigating the origin and composition of the at-leg pellet is probably a straightforward aim, but what made the authors think this structure might harbour a symbiotic fungus? Also, it is essential to say that these at-pellet may appear on several legs on the same insects.
4. Lines 69 and 86: Avoid the phrasing “according to previous studies” if only one study is referenced. Please reword these sections.
5. Line 69: provide more information/context for these tomato secretions. How are these produced? Function? Representation within the Solanaceae family? This would allow the reader to understand better why these secretions are not produced in cotton plants (line 75).
6. Line 103: Did the plates used to isolate the fungus only contain PDA? This is unusual, as PDA is a very rich medium and usually, isolation of microbes is cumbersome with many bacterial and fungal species being isolated. Also, state what PDA is.
7. Line 119 – I suggest the author refrain from interpreting/analysing the data in the Result section. The Result section should provide the facts in an unbiased manner.
8. Line 128: *P. citrinum* does not inoculate new mealybugs; it is transmitted horizontally to other insects via dispersion through the honeydew.

9. What is the source of *P. citrinum*? Could the authors provide a hypothesis? Clearly, the fungus uses the honeydew to grow and disperse through the plant and the mealybugs. However, plants with no mealybugs did not show fungal growth in the honey dew. One could then think that the insect is the source, but *P. citrinum* could not be isolated from insects with no at-leg pellets. This puzzles me. Is the fungus present in such small quantities in the tomato plants with no mealybugs (and therefore no honeydew to multiply) that cannot be re-isolated on PDA plates? Alternatively, the source of the fungus could be the insect, but why can it not be isolated from the insect – PDA is rich enough to provide enough fungal growth from a handful of spores.
10. What mechanism drives the antagonism between *P. citrinum* and *Akanthomyces* (*Verticillium*) *lecanii*. This hasn't been discussed at all. Is this competition for space or antibiosis?
11. Lines 174-175: 2.4 DTBP has a strong inhibitory effect on *Beauveria bassiana* and *A. lecanii*. The effect is so dramatic that prevents growth on the whole plate. Is the dose tested similar in plants and the wax of the at-leg pellet? 0.1 mg/ml is relatively high. Line 448, probably 30 ul and not 30 ml of the compound, was applied to the disc.
12. Line 188: Do the glandular trichomes produce the tomato secretions as a response to trichome damage?
13. Line 192-197: reduced expression of proteinase inhibitor 2 (PIN2) could result from mealybugs excreting more protein effectors into the plant as they tend to feed more. This hypothesis is presented in the Discussion, but in the Result section, the authors state that reduction of PIN2 expression is a direct consequence of the presence of the at-leg pellet. Correlation is not causation. Again, avoid interpreting the data within the Results section. Also, state what PIN2 is.
14. Provide more context in the Methodthods about PIN2.
15. Line 461: Effects of pellets on the plant antiherbivore defensive response is more adequate.
16. Line 204: "suggesting that insects with pellets and no pellets might have different feeding strategies." This is vague and should be adequately discussed in the Discussion.
17. Line 230-231: It is an adaptive response to overcome....
18. Overall, the Discussion would benefit from a more in-depth analysis. Some statements are superficial and provide limited context of the system examined. In addition, there aren't any references to key or similar studies linked to the topic.

Reviewer #3 (Remarks to the Author):

Mutualistic interactions between insects and fungi are widespread in nature, although they have only been documented in detail in the context of several insect-fungal systems. In the study reviewed herein, the authors describe a novel association between mealybugs, of particular interest as they constitute major plant pests, and *Penicillium citrinum*, a fungus known to occur in a wide variety of habitats, but to date unknown in the context of insect mutualisms. The authors demonstrated that insect-derived waxy filaments and host plant-derived secretions contribute to the formation of at-leg pellets, and that these pellets appear to be exclusively colonized by a *P. citrinum* isolate. Through a series of experiments, these pellets and their associated fungus were demonstrated to have antimicrobial properties, to protect mealybugs from pathogens in the environment, to alter insect feeding behavior, and to modulate host plant defense responses. Furthermore, *P. citrinum* was shown to colonize mealybug-derived honeydew

on leaf surfaces, where it could outcompete fungal pathogens of mealybugs and act as an environmental reservoir to colonize new mealybugs. In this sense, mealybugs may provide an enticing environment (honeydew) in which *P. citrinum* can grow and the presence of this fungus, along with host plant- and insect-derived substances, can exert positive effects on mealybug physiology and survival on host-plants. Overall, this manuscript provides an interesting and detailed examination of a previously undescribed insect-fungal-plant system that broadens the scope of knowledge regarding both mealybug and *P. citrinum* biology. Both experimental design and statistical analyses applied to the results seem relatively sound, and I have mostly only minor comments regarding the extrapolation of results from *in vitro* experiments to an *in vivo* system, and of the commonality of these findings in nature when they were only demonstrated on one host plant species, despite being demonstrated in several species of mealybug. The manuscript was well written, and the authors took time and effort to thoroughly examine this system from a number of angles (insect behavior, plant immunity, niche competition, pathogen protection) and to demonstrate their findings through a series of both *in vitro* and *in vivo* experiments. This study is likely to influence the field of insect-fungal interactions by contributing to the number of known insect-fungal mutualisms and to the mechanistic bases of their physical and chemical foundations. I look forward to following the work of these authors in the future and seeing how they will continue to build on the findings presented here. I think it would be particularly interesting to look at pellet formation on mealybugs feeding on other host plants which produce secretions, and whether, in these systems, the associated fungus is still a *P. citrinum* strain, or if other fungi would be uncovered. This could shed light on the flexibility and context dependency of this interaction, and whether numerous fungal species or strains could interact with mealybugs to fill this mutualistic niche. Please find below my suggestions and comments for revision of this manuscript.

1) Line 24: remove the word “the” for better sentence flow.

2) Line 39: change “community” to communities.

3) Line 65: Do the host plants of these other mealybugs also produce secretions? Have pellets been observed for these mealybug species on their usual host plants, or just on tomato?

4) Line 68: It is suggested that at-leg pellets may be common in mealybugs, however in this study they were only observed on one plant host. Is it possible to observe the presence/absence of these pellets and associated fungi on field collected specimens of mealybugs from other plant hosts that are known to produce secretions?

5) Line 106: Although pellet-free legs did not yield any consistent fungal isolate, was *P. citrinum* still cultured from these legs occasionally among other fungi, or were no fungal isolates observed from these samples? I am a little unclear on the wording in this statement. The figure (figure 3C) shows no fungi isolated from pellet-free legs, but I am a little surprised that these legs would be completely free of any fungus. It would be interesting to see if this finding holds up under field conditions, in addition to the laboratory experiments shown here.

6) Line 122 and Figure 3: In this section of text, the terms “*Penicillium citrinum* Fermentation product” and “Butylparaben” were introduced and later referred to in Figure 3 by the abbreviations “PCF” and “BP”, respectively. Although these abbreviations were explained in the figure legend, I feel that it may also be helpful to define them in the original section of text where they are first introduced.

7) Figure 4i: Remove the word “in” from the y-axis label.

8) Line 155: Change the word “invested” to “investigated”.

9) Line 160: Although these experiments convincingly demonstrate that *P. citrinum* can outcompete *V. lecanii* in an *in vitro* coculture, I would exercise some caution in stating that this ability is necessarily due

to its antimicrobial properties, as you did not demonstrate that directly in this assay. Properties such as relative growth rates in this medium could also result in one species outcompeting another. The *in vitro* conditions (liquid culture in a shaker incubator) could also influence the outcome of these species' relative growth rates as compared to their potential interactions in honeydew on a plant surface.

10) Figure 5: The 2,4-DTBP peaks in Figures 5A (at-leg pellets) and 5B (tomato leaves) both appear to have comparable retention times (~8.4 minutes) while the peak labeled 2,4-DTBP from *Penicillium citrinum* fermentation products (Figure 5C) appears to elute a full 1 minute earlier. Is this because the samples were prepared in different solvents (hexane vs. ethyl acetate), or is there some other aspect of the analysis that I am missing?

11) Line 178: Since you performed inhibition experiments with both *V. lecanii* and *B. bassiana in vitro*, it would have been interesting to also include *B. bassiana* in these *in vivo* protection experiments.

12) Line 185: It might be interesting to look at another metric for plant defense activation in addition to *PIN2* expression, such as the accumulation of defensive compounds in the leaves following exposure to pellet-containing or pellet-less mealybugs.

13) Line 188: Add the word "to" before the words "glandular trichomes".

14) Line 211: Remove the word "The" from the beginning of the sentence.

15) Line 219: Secretions of potatoes are mentioned in this sentence, but potatoes were previously mentioned to not produce much secretion, and mealybugs feeding on this host plant did not form pellets. It would be nice to elaborate on this discrepancy a little more.

16) Line 224: Are you suggesting that mealybugs redirect these secretions from their mouthparts as well? What is the fate of secretions accumulating on the mouthparts of mealybugs feeding on tomato?

17) Line 230: I think in order to call this adaptation common, it may be important to show that it occurs on more than one host plant. It is great that you could demonstrate pellet formation for multiple mealybug species, but I think only showing it on one host plant is a potential limitation of this study.

18) Line 246: Pluralize "tarsus" (tarsi).

19) Line 259: Change the word "outcompletes" to outcompetes.

20) Line 264: Add the word "in" before "*in vitro*".

21) Line 274: Change the word "fungi" to "fungus".

22) Line 293: Change 2rd to 2nd

23) Line 493: Add the word "with" after "mealybugs".

24) Line 494: Change "mealybugs" to "mealybug".

25) Lines 503-504: Change "it probes" to "they probe".

Point to point responses to reviewers

Reviewer #1 (Remarks to the Author):

In this manuscript Li and colleagues describe the presence of an at-leg pellet in mealybugs of the species *Phenacoccus solenopsis* growing on tomato plants. This pellet, while originated from host secretions and waxy filaments, also contains a fungal strain: *Penicillium citrinum*. The authors explore the role that these pellets and associated fungi play in (a) defending mealybugs and tomato plants from pathogens, (b) regulating mealybug behavior and (c) host defenses, and (d) colonizing new insect hosts.

This is a fascinating and timely discovery given that fungi are increasingly being recognized as major players in the ecology of insects.

While I enjoyed reading this manuscript, I have major concerns regarding the methodology employed to conclude that the *P. citrinus* strain isolated from mealybugs produces antimicrobial compounds that protect mealybugs against pathogens (see comments below). Likewise, the authors do not mention whether the antimicrobial compounds presumably produced by the fungus are synthesized in the pellet. Therefore, the potential compounds of fungal origin, may not be of ecological relevance.

Response:

We greatly appreciate your professional comments. We considered seriously your comments during revision, and each of methodological concerns has been addressed. Also, we have performed the experiment to clarify whether the antimicrobial compounds produced by the fungus can be synthesized in the pellet.

The corresponding modifications made to the text are listed below, and in the text they are highlighted in yellow.

- 1) Adding the specific rearing conditions of the insects (Q12 on Page 3-4, Lines 330-333);
- 2) Explaining the justification of clones for PCR product (Q14 on Page 4, Lines 399-401);
- 3) Describing the detailed procedures of antimicrobial assays on rice extract (Q15 on Page 4, Lines 134-136 and 418-424), and found that it does not have antimicrobial properties against the microorganisms under tests;
- 4) Giving the method (LC-MS) of investigating butylparaben and citrinin in rice, and we found that butylparaben, one of the dominant antimicrobial compounds produced by *P. citrinum*, is really synthesized in the pellet, but citrinin, another dominant compound, is not (Q10 on Page 3, Lines 197-199). And none of them were found in RE (Q16 on Page 4-5, Lines 140-141). This suggests that butylparaben is ecologically important for mealybugs (Q10, on Page 3, Lines 197-199); and
- 5) Giving details of the observation of the cause of mealybug mortality as well as the justification of the methodology (Q17 on Page 5, Lines 523-525 and 526-527).

The following is my list of revisions. If there are any incorrect or unclear points in the manuscript, please do not hesitate to let us know.

Q1: L37: I am not aware of any example of fungal defense against parasitoids, and I am not sure the cited work features any. The authors may consider verifying this. However, recent relevant studies are missing from the examples such as fungi protecting beetle pupae from predation (Berasategui et al. 2022).

Response:

Many thanks for pointing out this problem. We have added Weis (1982) as a supporting reference, where the author stated that the symbiotic fungus *Sclerotium asteris* of an gall midge (*Asteromyia carbonifera*) forms stromata to exclude the parasitoid (*Torymus capite*) of the insect (Line 38). With this reference we verified the “fungal defense against parasitoids”.

We have also cited some recent relevant studies in the text, including Biedermann and Vega (2020), and Berasategui et al. (2022), as examples of the protections conferred by fungi to natural enemies (Lines 37-38).

References:

- Weis, E. Use of a symbiotic fungus by the gall maker *Asteromyia carbonifera* to inhibit attack by the parasitoid *Torymus capite*. *Ecology* 63, 1602-1605 (1982).
- Biedermann, P.H.W., Vega, F.E. Ecology and evolution of insect-fungus mutualisms. *Annu. Rev. Entomol.* 65, 431-455 (2020).
- Berasategui, A., et al. The leaf beetle *Chelymorpha alternans* propagates a plant pathogen in exchange for pupal

protection. *Curr. Biol.* 32, 4114-4127 (2022).

Q2: L49: what type of typical insect adaptations are missing in mealybugs?

Response:

We have modified this sentence. Instead of depicting “missed typical adaptations” in mealybugs, we turn to describe some of their key life features (i.e., wingless and living a sedentary life) and the approaches to adapting to environments (Lines 49-52).

Q3: L63: If Figure 1 is to be a graphical abstract, then Fig. 2a, should just be Fig. 1a...and so on.

Response:

Good idea! Thanks. The numbers of figures have been adjusted correspondingly.

Q4: L65: It would be helpful to use an arrow to point to the pellet in Figure 2d. Pellets are generally difficult to observe in these photographs.

Response:

We’re sorry for our negligence. We have used yellow arrows there. By the way, Fig. 1d (Original Figure.2d) actually shows the leg of a newly emerged adult cotton mealybug female that does not have pellets.

Q5: L91: I find this result very intriguing/interesting. Why would the second generation have more at-leg pellets than the first one? What processes would be driving this phenomenon? Authors could speculate.

Response:

The increase of pellet number along with insect generations is probably correlated to the increase of glandular trichomes’ density and their secretions as plants grow (Saeidi *et al.*, 2007). We have stated this point in Lines 88-90.

Reference:

Saeidi, Z., Mallik, B., Kulkarni, R.S. Inheritance of glandular trichomes and two-spotted spider mite resistance in cross *Lycopersicon esculentum* "Nandi" and *L. pennellii* "LA2963". *Euphytica* 153, 231-238 (2007).

Q6: L96: I agree with the authors that these results do not demonstrate the negative effect of the pellets on insect performance. This is mainly because the experiment was not set up in a way that could answer that question. Perhaps the authors should consider changing the logic behind this experiment as stated in L86-87.

Response:

As the reviewers/editor commented that the text contains many overstatements and this should be avoided, and the topic of this section is “Characterization and source of pellets on mealybug legs”, we have deleted the statement about “negative effect” of pellets and meanwhile reduced the words for pellets in 2nd generation. Thus, in the revised version, the original 3rd paragraph (with the topic “whether pellets negatively impact mealybugs”) has been deleted, and the information of pellets in 2nd generation has been combined into 1st paragraph (please see the highlighted Lines 88-90).

Q7: L104: What is the similarity percentage between the isolated fungal strain to *P. citrinum* from the database?

Response:

The similarity is 100% with the *P. citrinum* from database (accession ID: KX664347, reported by Mayer *et al.*, 2016), based on the comparison of a 543-bp fragment. Such information has been added in Lines 115-116.

Reference:

Mayer, T., *et al.* Microbial succession in an inflated lunar/Mars analog habitat during a 30-day human occupation. *Microbiome* 4, 22; 10.1186/s40168-016-0167-0 (2016).

Q8: L107: Did the pellet-free legs failed to produce consistent fungal morphologies, or no fungal isolates at all?

Response:

We’re sorry for having not described the result clearly. We meant the pellet-free legs could produce a few kinds of fungal isolates that differed in morphology, and the type of isolates could also differ with trials (Lines

116-120).

By the way, similar questions were also raised by the other two reviewers. For details, please refer to our responses to the comment Q5 of the Reviewer #3 on Page 13-14 of this document.

Q9: L160: Given that *P. citrinum* grows better in a mixture with *V. lecanii* than alone, this also suggests that *P. citrinum* is benefiting off *V. lecanii* in some way. Authors could speculate as to why.

Response:

Thank you very much for your suggestion. We have arranged one paragraph (the 5th one in Discussion, Lines 287-296) to speculate the ways/reasons of “*P. citrinum* is benefiting off *V. lecanii*”, from the points of antibiosis and the shift of reproductive strategy in *P. citrinum*. The content of the paragraph is as below.

“In honeydew, *P. citrinum* grows better in a mixture with *A. lecanii* than alone, while *A. lecanii* grows worse (Fig. 3j), which suggests that *P. citrinum* could have benefited from *A. lecanii* in some ways. There are two possible reasons for this phenomenon. One is antibiosis. In other words, by secreting certain compounds, such as butylparaben, *P. citrinum* may have inhibited the growth of *A. lecanii* to certain degree (Fig. 2k). The other reason is the shift of *P. citrinum* reproductive strategy as meeting a more limited resource. According to Chan *et al.* (2019) and Gilchrist *et al.* (2006), as the resource available depletes, fungi tend to allocate more energy to reproduction. Thus, when coexisting with *A. lecanii* in the honeydew, *P. citrinum* might have reproduced more rapidly than existing alone, as a response to a rapider consumption of available resource.”

Q10: L166: Were the compounds isolated from the fermentation of *P. citrinum* (citrinin and butylparaben) also found in these extracts? If not, these compounds may not be produced in natural conditions, and may therefore not be ecologically relevant.

Response:

Yes, butylparaben (of antimicrobial ability) was found in pellets, but citrinin was not (see Fig. 2i, j below and Line 197), which are two of the dominant components of *P. citrinum* fermentation products (Supplementary Table 1). This suggests that at least butylparaben can be produced by *P. citrinum* in natural conditions and be included in pellets, thereby playing ecological roles during interactions of *P. citrinum* or pellets with other fungi.

In addition, we explained partially why butylparaben can be detected in the pellets but citrinin cannot (Lines 271-276).

The butylparaben (i) and citrinin (j) detected in pellets and rice extract in the LC-MS analysis (From Fig. 2: Morphology and antimicrobial properties of *Penicillium citrinum* identified from pellets).

Q11: L170: Is Fig2c correct?

Response:

“Fig. 2c” should not be placed on this line. It has been deleted.

Q12: L280-285: How were the insects reared? (Use of cages, potted plants etc)

Response:

The insects were reared in cages (40 cm × 50 cm × 50 cm) with potted tomato.

We have adjusted the statement to “The collected insects were reared in screened cages (40 cm × 50 cm × 50 cm) at the Zijingang Campus of Zhejiang University (ZU) with potted tomato and cotton for 2-4 generations prior to use in a phytotron maintained at 26 ± 1°C and RH 70 ± 5% with a photoperiod of 14 h : 10 h (L : D)” in

Materials and methods (Lines 330-333).

Q13: L336: What were the growing conditions?

Response:

All the plates were incubated at $27 \pm 1^\circ\text{C}$ and checked for fungal isolates after 72 h. Words have been added to Lines 385-388.

Q14:L350: Was the PCR product cloned because there were several ITS copies in the fungal genome?

Response:

Yes, because there may be more than one ITS copies in some fungal genomes (Woo *et al.*, 2010), we used the cloning methods. Please see Lines 399-401.

Reference:

Woo, P.C.Y., *et al.* Internal transcribed spacer region sequence heterogeneity in *Rhizopus microsporus*: Implications for molecular diagnosis in clinical microbiology laboratories. *J. Clin. Microbiol.* 48, 208-214 (2010).

Q15: L362: I have concerns regarding the antimicrobial assays. If I understood it correctly, the authors inoculated the isolated fungal strain onto liquid media, and then this media into sterilized rice. This mixture was then extracted with EtOAc. Treatment disks were generated by dipping disks into this extract mixture. However, control disks were generated by dipping them in EtOAc alone. This suggests that the treatment disks will contain secondary metabolites originating from both the PDA liquid media, the fungal isolate and the rice, whereas the control treatment contains no secondary metabolites. If this is correct, this experiment is unfortunately flawed. The correct control would have been disks dipped in EtOAc extracts originating from PDA liquid media-inoculated rice. Perhaps I misunderstood the methodology, in which case I encourage authors to clarify the procedure.

Response:

It's a great comment that reminds us of paying attention to potential effects of the secondary metabolites originating from PDA liquid media and rice. Actually, before assessing *P. citrinum* fermentation products, we had previously assayed the antimicrobial ability of rice extract (RE) + PDA liquid media against the fungi; the results showed that they did not have such ability to each of the four fungal/bacterial strains tested.

We have modified the sentence on Lines 134-136 and 418-423, and provided Supplementary Fig. 4 as given below.

Supplementary Figure. 4 Antimicrobial ability assays of rice extract against *Akantaomyces lecanii* (a), *Beauveria bassiana* (b), *Pseudomonas syringae* (c) and *Botrytis cinerea* (d) using the disk-diffusion method.

EtOAc: Ethyl acetate; RE: rice extract containing PDA liquid medium.

Q16: L371: In the same manner, I have concerns about the experiment exploring the origin of the antimicrobial activity observed. Control extracts were not analyzed, and hence, it is unclear whether the detected compounds butylparaben and citrinin originated from the fungus or rice. In fact, butylparaben has been detected in grains such as barley and flax seed, and is usually employed in industry for its antimicrobial (both antifungal and antibacterial) properties. Citrinin, is indeed produced by some fungi and is known to have antimicrobial properties, but in these experiments, it fails to inhibit pathogens. Additionally, it would be important to explicitly mention that commercial standards were used to perform the antimicrobial assays.

Response:

We have performed a supplementary detection to determine whether the control extracts (rice extract + PDA liquid media) contain butylparaben and citrinin, and none of them were found (Fig. 2i, j; Lines 140-141). Therefore, the detected compounds butylparaben and citrinin originated from *P. citrinum*, but not from rice. In other words, the observed antimicrobial activity originated from the fungi only. This is accordant with the result of assays described above that rice extract (RE) + PDA liquid media did not have antimicrobial ability (also see our reply to Q15).

We're also surprised that in our experiments citrinin failed to inhibit pathogens. The reason is not clear. We have also mentioned this phenomenon again in the Discussion for readers' attention: "However, in our study, citrinin did not display antimicrobial ability against *A. lecanii* and *P. syringae*. In another report, citrinin was stated as the defense metabolite of *Penicillium corylophilum* stressed with the antagonist fungus *B. bassiana* (Campinha dos Santos *et al.*, 2012), certain strains of which are pathogens of mealybugs (Amnuaykanjanasin *et al.*, 2013; Tomson *et al.*, 2021). Thus, citrinin holds the potential of protecting mealybugs from pathogens, though our study failed to demonstrate this and the responsible reasons are not clear." (Lines 257-263).

All antimicrobial assays referred to the standards set by Clinical and Laboratory Standards Institute (CLSI, 2010; 2021). This information has been given on Lines 423-424.

References:

- Clinical and Laboratory Standards Institute. Method for antifungal disk diffusion susceptibility testing of nondermatophyte filamentous fungi. M51-A (Clinical and Laboratory Standards Institute, 2010).
- Clinical and Laboratory Standards Institute. Performance Standards for antimicrobial susceptibility testing. 31st ed (Clinical and Laboratory Standards Institute, 2021).

Q17: L450: Mealybugs were considered killed by V. lecanii if dead animals placed on PDA media grew the pathogen. However, authors do not differentiate between animals that died with V. lecanii vs. those that died because V. lecanii. An animal with one spore of V. lecanii on the surface, that died for a different reason, would still be V. lecanii positive on a plate and recorded as killed by the pathogen. Perhaps measuring pathogen abundance via qPCR would give a better estimate.

Response:

We agree with this comment that even if there is only one spore on the surface of dead insect, there would be growing of *A. lecanii* on plate. However, we think such a possibility would not influence our results, because:

- 1) If the insects that died for other reasons have *A. lecanii* spores on body surface could produce *A. lecanii* colony (colonies) on plates, then it could be suspected that the treatment (i.e., surface sanitized using 0.6% sodium hypochlorite and 75% ethanol) is not as good as expected and then we would probably find other fungi on plates besides *A. lecanii*. But, we rarely meet such cases during assays, and the few dead insects in such cases were not taken into account, as stated in our text: "The mealybugs were recorded as dead due to *A. lecanii* infection only when the single fungus *A. lecanii* colonized the PDA, as could be identified according to their morphology" (Lines 526-527).
- 2) Surface sanitization using 0.6% sodium hypochlorite and 75% ethanol has been testified to be rather effective (Lacey and Brooks, 1997) and frequently adopted by other researches to determine whether the insects died of pathogens or not (Traniello *et al.*, 2002) (Lines 523-525).

References:

- Lacey, L.A., Brooks, W.M. Manual of techniques in insect pathology 1-16 (Academic Press, 1997).
- Traniello, J.F.A., Rosengaus, R.B., Savoie, K. The development of immunity in a social insect: Evidence for the group facilitation of disease resistance. *Proc. Natl. Acad. Sci. U. S. A.* 99, 6838-6842 (2002).

Q18: L209: Correct citation: Biedermann and Vega.

Response:

Done as suggested (Line 229).

Q19: L223: I disagree with the idea that the at-leg pellets explain the mechanism by which mealybugs avoid lethal tomato secretions. Even if the animals redirect plant secretions for the formation of the pellets, these could still be

toxic. Authors could consider rephrasing.

Response:

Thanks for your suggestions. During revision, we have excluded the discussion about contributions of pellets to avoidance of lethal tomato secretions.

Q20: L230: I suggest authors to not to overstate the role of at-leg pellets in mealybug overcoming of host plant secretions.

Response:

Thanks again for your comment. As we reply to Q19, such discussion has been deleted in the text.

Reviewer #2 (Remarks to the Author):

In this manuscript, Li et al. report the formation of an at-leg pellet derived from tomato secretions and stores the fungus *Penicillium citrinum*. This pellet appears to be an adaptive response of the mealybug to potential trichome toxic secretions and provides a reservoir of the fungus *P. citrinum*, which can protect against fungal diseases in the mealybug. Overall, the paper is well written, the experiments are solid, and the results are clearly presented and discussed. I have some minor comments that could potentially improve the manuscript.

Response:

We really appreciate the insightful comments on our manuscript. We have made modifications according to the suggestions and included point-by-point responses to each issue with the revisions highlighted in yellow.

*Q1. The currently accepted name for *Lecanicillium lecanii* is *Akantaomyces lecanii* (Zimmerman) (species fungorum). Correct this throughout the manuscript.*

Response:

Thank you for this crucial point. All of the “*Lecanicillium lecanii*” (*L. lecanii*) have been replaced with “*Akantaomyces lecanii*” or “*A. lecanii*” throughout the manuscript.

By the way, such replacements have been made directly in the text (not in revision status), as there are so many “*L. lecanii*” in the original manuscript.

Q2. Line 47-49 – The two sentences appear to be incomplete. “Causing losses” on what? And “lack many typical adaptations.” these sentences need additional context.

Response:

We have modified the original sentence as “damage crops and landscape trees, causing losses in yield or ornamental value (Waqas *et al.*, 2021)” (Lines 48-49). And instead of depicting “missed typical adaptations” in mealybugs, we turn to describe some of their key life features (i.e., wingless and living a sedentary life) and the approaches to adapt to environments (Lines 49-52).

Q3. It would be good if the authors provided more information about the at-leg pellet in the introduction. This would give a better justification for the study. Indeed, the introduction doesn't offer any rationale and/or hypotheses. Why is it important to study the function of this “organ.” Are there similar structures formed in other insects? What background information exists about at-leg pellets in mealybugs? Have these at-leg pellets been observed recently? Investigating the origin and composition of the at-leg pellet is probably a straightforward aim, but what made the authors think this structure might harbour a symbiotic fungus? Also, it is essential to say that these at-pellet may appear on several legs on the same insects.

Response:

Thanks for your valuable comments. We have added two paragraphs in ‘Introduction’, where all of the aspects you suggested have been included there (Lines 53-73).

Q4. Lines 69 and 86: Avoid the phrasing “according to previous studies” if only one study is referenced. Please reword these sections.

Response:

Thanks for your comment. We have corrected the singular or plural forms of nouns throughout the manuscript.

Q5. Line 69: provide more information/context for these tomato secretions. How are these produced? Function? Representation within the Solanaceae family? This would allow the reader to understand better why these secretions are not produced in cotton plants (line 75).

Response:

Great comments! We have provided such information and context in the text, including (see the underlined words as key information)

- 1) Lines 55-58 in Introduction: “It surprised us, because in some other insects such as leafhopper and aphids, the substances (secretions of glandular trichomes) attached or coated onto their legs during feeding on *Solanum* plants would impact their life and even cause death due to the impediment of movement and feeding (Tingey

and Gibson, 1978; Steffens and Walters, 1991).”

- 2) **Lines 68-71** in Introduction: “The source of such fungal spores, if they are confirmed to be, also interests us, because normally secretions of tomato glandular trichomes normally have antimicrobial activity (Zabel *et al.*, 2021) and are probably unfavorable for establishment of most fungi on tomato.”
- 3) **Lines 88-90** in Results: “As mealybugs developed into the next generation on tomato, during which the density of glandular trichomes and their secretions on plants increased along (Saeidi *et al.*, 2007), both percentage and number of pellets increased significantly (Fig. 1e, f) ($P < 0.05$).”
- 4) **Lines 100-104** in Results: “We also observed the situation on cotton (*Gossypium hirsutum*) that has little plant secretions on leaves or stems, and on potato (*Solanum tuberosum*) and eggplant (*Solanum melongena*) that have fewer types of trichomes relative to tomato (Glas *et al.*, 2012; Tai *et al.*, 2014; Cho *et al.*, 2017; Jayanthi *et al.*, 2018).”
- 5) **Lines 231-233** in Discussion: “Particularly, the pellets are formed by mealybugs in such a kind of plants (*Solanum*) that typically use the secretions from glandular trichomes to defend herbivory insects (Riddick and Simmons, 2014).”
- 6) **Lines 237-241** in Discussion: “As shown in our study, *P. solenopsis* did not form pellets at their legs on potato and eggplant (Supplementary Fig. 1), which have fewer types of glandular trichomes relative to tomato (Glas *et al.*, 2012; Cho *et al.*, 2017; Jayanthi *et al.*, 2018). Different types of glandular trichomes have been known to generate different kinds and amount of secretions (Glas *et al.*, 2012).”

Q6. Line 103: Did the plates used to isolate the fungus only contain PDA? This is unusual, as PDA is a very rich medium and usually, isolation of microbes is cumbersome with many bacterial and fungal species being isolated. Also, state what PDA is.

Response:

Yes, in the previous experiments, we used only PDA plates to isolate fungi. To find more fungi possibly present in pellets, we performed additional isolations using plates of Czapek dox agar (CzA), yeast extract peptone dextrose medium (YPD) and Sabouraud dextrose agar (SDA). Again, only *P. citrinum* was isolated. Please see **Lines 121-125** and for the result of isolations using these plates, and **Lines 385-388** for corresponding methods.

We have extended the description of PDA as “potato dextrose agar” (**Lines 110-111**).

Q7. Line 119 – I suggest the author refrain from interpreting/analysing the data in the Result section. The Result section should provide the facts in an unbiased manner.

Response:

We have done as you suggested: already deleting the original sentence on Lines 118-119, and checking through the Result section for this problem.

Q8. Line 128: P. citrinum does not inoculate new mealybugs; it is transmitted horizontally to other insects via dispersion through the honeydew.

Response:

Thanks for your suggestion! The subtitle “*P. citrinum* colonized, proliferated, and inoculated new mealybugs in mealybug honeydew” has been modified to “Colonization and proliferation of *P. citrinum* in honeydew and horizontal transmission to mealybugs” (**Lines 144-145**).

Q9. What is the source of P. citrinum? Could the authors provide a hypothesis? Clearly, the fungus uses the honeydew to grow and disperse through the plant and the mealybugs. However, plants with no mealybugs did not show fungal growth in the honey dew. One could then think that the insect is the source, but P. citrinum could not be isolated from insects with no at-leg pellets. This puzzles me. Is the fungus present in such small quantities in the tomato plants with no mealybugs (and therefore no honeydew to multiply) that cannot be re-isolated on PDA plates? Alternatively, the source of the fungus could be the insect, but why can it not be isolated from the insect – PDA is rich enough to provide enough fungal growth from a handful of spores.

Response:

Good question! We hypothesized that, for the *P. citrinum* in honeydew, pellets (carrying *P. citrinum* spores) are the source, and for the *P. citrinum* spores in pellets, the fungus in honeydew is the source, because the fungus can be transmitted from pellets to honeydew and *vice versa*.

The above ideas have been demonstrated in Discussion, including:

- 1) **Lines 237-241**: “As shown in our study, *P. solenopsis* did not form pellets at their legs on potato and eggplant (Supplementary Fig. 1), which have fewer types of glandular trichomes relative to tomato (Glas *et al.*, 2012; Cho *et al.*, 2017; Jayanthi *et al.*, 2018). Different types of glandular trichomes have been known to generate different kinds and amount of secretions (Glas *et al.*, 2012)” and
- 2) **Lines 279-284**: “Previous reports showed that honeydew of mealybugs may serve as substrates for the growth of their symbiotic fungi (Fang *et al.*, 2020; Gavrilov-Zimin, 2017). Similar results were obtained in our study (Fig. 3a-d), and we also found that the *P. citrinum* in honeydew can be transmitted to mealybugs, and the honeydew serves as an arena for *P. citrinum* interacting with microorganisms (e.g., pathogens of mealybugs and plants).”

It is possible that “the fungus present in such small quantities in the tomato plants with no mealybugs (and therefore no honeydew to multiply) that cannot be re-isolated on PDA plates” (**Lines 266-268**). And, such a possibility also exist for the fungus on insects with no at-leg pellets. Furthermore, we confirmed this issue by using other kinds of plates, i.e., CzA, YPD and SDA, and no *P. citrinum* was obtained either from plants with no mealybugs, or from insects with no at-leg pellets (**Lines 161-164**). Therefore, the *P. citrinum* on such plants and insects, if any, contribute very little as sources of *P. citrinum* occurring in pellets and honeydew.

Q10. What mechanism drives the antagonism between P. citrinum and Akanthomyces (Verticillium) lecanii. This hasn't been discussed at all. Is this competition for space or antibiosis?

Response:

Thank you very much for your suggestion. This question is similar to **Q9 on Page 3** from Reviewer #1.

We have arranged one paragraph (the 5th one in Discussion, **Lines 287-296**) to speculate the ways/reasons of “*P. citrinum* is benefiting off *V. lecanii*”, from the points of antibiosis and the shift of reproductive strategy in *P. citrinum*. The content of the paragraph is as below:

“In honeydew, *P. citrinum* grows better in a mixture with *A. lecanii* than alone, while *A. lecanii* grows worse (Fig. 3j), which suggests that *P. citrinum* could have benefited from *A. lecanii* in some ways. There are two possible reasons for this phenomenon. One is antibiosis. In other words, by secreting certain compounds, such as butylparaben, *P. citrinum* may have inhibited the growth of *A. lecanii* to certain degree (Fig. 2k). The other reason is the shift of *P. citrinum* reproductive strategy as meeting a more limited resource. According to Chan *et al.* (2019) and Gilchrist *et al.* (2006), as the resource available depletes, fungi tend to allocate more energy to reproduction. Thus, when coexisting with *A. lecanii* in the honeydew, *P. citrinum* might have reproduced more rapidly than existing alone, as a response to a rapider consumption of available resource.”

Q11. Lines 174-175: 2,4 DTBP has a strong inhibitory effect on Beauveria bassiana and A. lecanii. The effect is so dramatic that prevents growth on the whole plate. Is the dose tested similar in plants and the wax of the at-leg pellet? 0.1 mg/ml is relatively high. Line 448, probably 30 ul and not 30 ml of the compound, was applied to the disc.

Response:

According to the supplementary detection, the dose of 2,4-DTBP in pellets was 0.087 ± 0.002 mg per gram of pellets (Supplementary Fig. 6a), which was similar to the dose (0.1 mg/mL) of this chemical used in the assays with *Beauveria bassiana* and *A. lecanii*. So, the dose (0.1 mg/mL) is in a reasonable range.

We have already tested the antimicrobial ability of 2,4-DTBP with other three doses, 0.08, 0.01 and 0.001 mg/mL. The result of using 0.08 mg/mL 2,4-DTBP is similar, which strongly inhibits *A. lecanii* and *B. bassiana*, but we did not find such ability using 0.01 and 0.001 mg/mL (Fig. 4d, e; Supplementary Fig. 6b, c). These results have been integrated to **Lines 192-195**.

Regarding “probably 30 μ L and not 30 mL of the compound, was applied to the disc”, we’re sorry for not having described this step clearly. In this experiment, we first immersed the disks in 30 mL of 2,4-DTBP (0.1 mg/mL) that were contained in a tube, then took out the saturated disk; actually ca. 15 μ L of solution were infused in each disk. To avoid misunderstanding, the statement has been modified to “The disks were saturated separately with 0.08, 0.1, 0.01 and 0.001 mg/mL 2,4-DTBP dissolved in DMSO, using disks saturated with DMSO as control.” (**Lines 509-511**).

Q12. Line 188: Do the glandular trichomes produce the tomato secretions as a response to trichome damage?

Response:

We do not know whether the glandular trichomes of tomato produce secretions upon physical damage by mealybugs. However, it was reported that type VI of tomato glandular trichomes are ready to release secretions if given physical contact (Steffens and Walters, 1991; McDowell *et al.*, 2011). To reveal this point, some words have been added to the sentence in Lines 209-213, as listed below:

“Crawling by *P. solenopsis* caused physical damage to glandular trichomes on tomato leaves, irrespective of pellet presence or not (Supplementary Fig. 8), which, according to Steffens and Walters (1991) and McDowell *et al.* (2011), would likely induce the release of secretions from trichomes. However, ...”

References:

- Steffens, J.C., Walters, D.S. Biochemical aspects of glandular trichome-mediated insect resistance in the Solanaceae. *ACS Symp. Ser.* 449, 136-149 (1991).
McDowell, E.T., *et al.* Comparative functional genomic analysis of *Solanum* glandular trichome types. *Plant Physiol.* 155, 524-539 (2011).

Q13. Line 192-197: reduced expression of proteinase inhibitor 2 (PIN2) could result from mealybugs excreting more protein effectors into the plant as they tend to feed more. This hypothesis is presented in the Discussion, but in the Result section, the authors state that reduction of PIN2 expression is a direct consequence of the presence of the at-leg pellet. Correlation is not causation. Again, avoid interpreting the data within the Results section. Also, state what PIN2 is.

Response:

Thanks for valuable comments! We agree with your point that expression level of *PIN2* is related to feeding, has also been reported previously (Zhang *et al.*, 2015). However, such a feeding-induced case could not take place in the insects in our study, because, during our experiment, the insects were not allowed to feeding or defecating when they were crawling (actually they have no chance of feeding during crawling). Moreover, crawling by insects has been proved to be able to induce *PIN2* expression in tomato (Peiffer *et al.*, 2009). Therefore, the observed reduction of *PIN2* expression resulted from mealybugs' crawling, but not from their feeding or the protein effectors. Please refer to Lines 213-219.

The results of *PIN2* expression have been rephrased more carefully (Lines 213-219), and we have avoided the interpretation of data within the Results section. In addition, the subtitle has been modified to “Pellets matter with the antiherbivore defense response of host plants” (Line 207).

The *PIN2* has been given more words (Lines 214-215).

References:

- Peiffer, M., Tooker, J.F., Luthe, D.S., Felton, G.W. Plants on early alert: glandular trichomes as sensors for insect herbivores. *New Phytol.* 184, 644-656 (2009).
Zhang, P.J., Huang, F., Zhang, J.M., Wei, J.N., Lu, Y.B. The mealybug *Phenacoccus solenopsis* suppresses plant defense responses by manipulating JA-SA crosstalk. *Sci. Rep.* 5, 9354; 10.1038/srep09354 (2015).

Q14. Provide more context in the Methods about PIN2.

Response:

We have extended the description of *PIN2* as “the expression level of *PIN2*, which is a highly inducible gene encoding the protease inhibitor 2, a representative defense protein against herbivorous arthropods (Green and Ryan, 1972; Chye *et al.*, 2006)” (Lines 553-555).

Q15. Line 461: Effects of pellets on the plant antiherbivore defensive response is more adequate.

Response:

Thanks! The subtitle has been modified to “Effects of pellets on the plant antiherbivore defensive response” (Line 529).

Q16. Line 204: “suggesting that insects with pellets and no pellets might have different feeding strategies.” This is

vague and should be adequately discussed in the Discussion.

Response:

We have deleted the sentence above in the Result section, and we think it would be better not discuss “feeding strategies”, because actually we did not carry any experiments involving “feeding strategies” in our study.

We discuss the reason why “presence of pellets at the leg may lead to an increase in sucking frequency”.
Please see Lines 297-308.

Q17. Line 230-231: It is an adaptive response to overcome....

Response:

This sentence has been deleted in the revised version.

Q18. Overall, the Discussion would benefit from a more in-depth analysis. Some statements are superficial and provide limited context of the system examined. In addition, there aren't any references to key or similar studies linked to the topic.

Response:

Thanks for your suggestions. We have discussed more deeply by adding paragraphs and sentences, which focus on:

- 1) Relations of pellet formation with host kinds (Lines 237-247; our response to comments Q3 of Reviewer #3 on Page 12-13);
- 2) Source of the *P. citrinum* in pellets (Lines 251-254 and Lines 279-284; response to comments Q9 of Reviewer #2 on Page 8-9);
- 3) Mechanisms for *P. citrinum* outcompeting *A. lecanii* in honeydew (Lines 287-296; response to comments Q10 of Reviewer #2 on Page 9); and
- 4) Possible relations of pellets with mealybugs' sucking (Lines 297-308; response to comments Q16 of Reviewer #2 on Page 10-11).

In addition, a number of references have been added accordingly.

Reviewer #3 (Remarks to the Author):

Mutualistic interactions between insects and fungi are widespread in nature, although they have only been documented in detail in the context of several insect-fungal systems. In the study reviewed herein, the authors describe a novel association between mealybugs, of particular interest as they constitute major plant pests, and *Penicillium citrinum*, a fungus known to occur in a wide variety of habitats, but to date unknown in the context of insect mutualisms. The authors demonstrated that insect-derived waxy filaments and host plant-derived secretions contribute to the formation of at-leg pellets, and that these pellets appear to be exclusively colonized by a *P. citrinum* isolate. Through a series of experiments, these pellets and their associated fungus were demonstrated to have antimicrobial properties, to protect mealybugs from pathogens in the environment, to alter insect feeding behavior, and to modulate host plant defense responses. Furthermore, *P. citrinum* was shown to colonize mealybug-derived honeydew on leaf surfaces, where it could outcompete fungal pathogens of mealybugs and act as an environmental reservoir to colonize new mealybugs. In this sense, mealybugs may provide an enticing environment (honeydew) in which *P. citrinum* can grow and the presence of this fungus, along with host plant- and insect-derived substances, can exert positive effects on mealybug physiology and survival on host-plants.

Overall, this manuscript provides an interesting and detailed examination of a previously undescribed insect-fungal-plant system that broadens the scope of knowledge regarding both mealybug and *P. citrinum* biology. Both experimental design and statistical analyses applied to the results seem relatively sound, and I have mostly only minor comments regarding the extrapolation of results from *in vitro* experiments to an *in vivo* system, and of the commonality of these findings in nature when they were only demonstrated on one host plant species, despite being demonstrated in several species of mealybug. The manuscript was well written, and the authors took time and effort to thoroughly examine this system from a number of angles (insect behavior, plant immunity, niche competition, pathogen protection) and to demonstrate their findings through a series of both *in vitro* and *in vivo* experiments. This study is likely to influence the field of insect-fungal interactions by contributing to the number of known insect-fungal mutualisms and to the mechanistic bases of their physical and chemical foundations. I look forward to following the work of these authors in the future and seeing how they will continue to build on the findings presented here. I think it would be particularly interesting to look at pellet formation on mealybugs feeding on other host plants which produce secretions, and whether, in these systems, the associated fungus is still a *P. citrinum* strain, or if other fungi would be uncovered. This could shed light on the flexibility and context dependency of this interaction, and whether numerous fungal species or strains could interact with mealybugs to fill this mutualistic niche. Please find below my suggestions and comments for revision of this manuscript.

Response:

We greatly appreciate the insightful comments on our manuscript and the opportunity to submit a revision. All modifications were highlighted in yellow.

Q1) Line 24: remove the word “the” for better sentence flow.

Response:

Done as suggested (Line 24). Thanks!

Q2) Line 39: change “community” to communities.

Response:

Done as suggested (Line 40).

Q3) Line 65: Do the host plants of these other mealybugs also produce secretions? Have pellets been observed for these mealybug species on their usual host plants, or just on tomato?

Response:

Yes, some host plants of these other mealybugs (*Phenacoccus solani* and *Paracoccus marginatus*) also produce secretions. Both mealybugs can use some species of Solanaceae as hosts, which, according to Ben-Dov (2005) and Macharia (2017), are frequently present with glandular trichomes (Lines 245-247).

However, we haven't observed the situation of their pellets on their usual host plants (observed just on tomato). To show the significance of such observations in the future, we have added some sentences as presented below (Lines 243-247):

“More efforts are needed to learn more cases of mealybugs carrying pellets on their secretion-producing hosts

(including wild and cultivated plants), for example, investigating the situation of *P. solani* on tobacco, which has glandular trichomes (Ben-Dov, 2005; Huchelmann *et al.*, 2017), and of *P. marginatus* on eggplant which also has trichomes (Macharia *et al.*, 2017; Jayanthi *et al.*, 2018).”

References:

- Ben-Dov, Y. The solanum mealybug, *Phenacoccus solani* ferris (Hemiptera: Coccoidea: Pseudococcidae), extends its distribution range in the Mediterranean basin. *Phytoparasitica* 33, 15-16 (2005).
- Macharia, I., *et al.* First report and distribution of the papaya mealybug, *Paracoccus marginatus*, in Kenya. *J. Agr. Urban Entomol.* 33, 142-150 (2017).

Q4) Line 68: It is suggested that at-leg pellets may be common in mealybugs, however in this study they were only observed on one plant host. Is it possible to observe the presence/absence of these pellets and associated fungi on field collected specimens of mealybugs from other plant hosts that are known to produce secretions?

Response:

Actually, before our study was conducted, we had frequently observed the presence of pellets on field collected *P. solenopsis* from tomato, but the pellets are absent on those collected from cultivated potato (Cho *et al.*, 2017), eggplant (Jayanthi *et al.*, 2018), pepper (Kim *et al.*, 2012) and tobacco (Huchelmann *et al.*, 2017), each of which is present with the trichomes that produce secretions. Thus, whether the pellets can be formed appears to be dependent on the density and type of glandular trichomes, as we discussed in the text (Lines 237-243). As Solanaceae plants, particularly those of *Solanum*, are often present with glandular trichomes, pellets would be hopefully found for the mealybugs on other hosts.

Yet, before concluding the phenomenon is common or not, a more extensive investigation is needed, as we stated in the text (Lines 243-247): “More efforts are needed to learn more cases of mealybugs carrying pellets on their secretion-producing hosts (including wild and cultivated plants), for example, investigating the situation of *P. solani* on tobacco, which has glandular trichomes (Ben-Dov, 2005; Huchelmann *et al.*, 2017), and of *P. marginatus* on eggplant which also has trichomes (Macharia *et al.*, 2017; Jayanthi *et al.*, 2018).”

We also have found *P. citrinum* in the pellets of field collected *P. solenopsis*. Please see Lines 121-125.

References:

- Cho, K.S., *et al.* Characterization of trichome morphology and aphid resistance in cultivated and wild species of potato. *Hortic. Environ. Biotechnol.* 58, 450-457 (2017).
- Huchelmann, A., Boutry, M., Hachez, C. Plant glandular trichomes: Natural cell factories of high biotechnological interest. *Plant Physiol.* 175, 6-22 (2017).
- Jayanthi, P.D.K., *et al.* Morphological diversity of trichomes and phytochemicals in wild and cultivated eggplant species. *Indian J. Hortic.* 75, 265-272 (2018).
- Kim, H.J., *et al.* Morphological classification of trichomes associated with possible biotic stress resistance in the Genus *Capsicum*. *Plant Pathol. J.* 28, 107-113 (2012).

Q5) Line 106: Although pellet-free legs did not yield any consistent fungal isolate, was *P. citrinum* still cultured from these legs occasionally among other fungi, or were no fungal isolates observed from these samples? I am a little unclear on the wording in this statement. The figure (figure 3C) shows no fungi isolated from pellet-free legs, but I am a little surprised that these legs would be completely free of any fungus. It would be interesting to see if this finding holds up under field conditions, in addition to the laboratory experiments shown here.

Response:

P. citrinum was not cultured from pellet-free legs. However, certain kinds of fungi were isolated occasionally from these legs: we used 30 pellet-free mealybugs for isolation, but only two of them produced fungal isolates, and only 1-2 isolates were isolated from pellet-free legs of each of these two mealybugs (Lines 116-120).

To obtain isolates that probably cannot be obtained using PDA (as suggested by Reviewer #2, Q6 on Page 8), in the last month we performed additional isolation using plates of Czapek dox agar (CzA), yeast extract peptone dextrose medium (YPD) and Sabouraud dextrose agar (SDA), for both legs with pellets and those without pellets of field collected mealybugs. Again, only *P. citrinum* was isolated from legs with pellets (Supplementary Fig. 3a, c, e) and no isolate was obtained from the legs without pellets (Supplementary Fig. 3b, d, f).

Regarding “It would be interesting to see if this finding holds up under field conditions, in addition to the

laboratory experiments shown here”: we tried to isolate other fungi with CzA, YPD and SDA plates, and again no fungi were generated from pellet-free legs of mealybugs that were collected from field (Lines 121-125).

Supplementary Figure. 3 Morphology of *P. citrinum* from field-collected mealybug’s legs on different medium. *P. citrinum* colony was generated from mealybug legs with pellets on Czapek dox agar (CzA) medium (a), Yeast extract peptone dextrose (YPD) medium (c), and Sabouraud dextrose agar (SDA) medium (e). No fungi were isolated from legs of mealybugs without pellets on CzA (b), YPD (d), and SDA (f).

Q6) Line 122 and Figure 3: In this section of text, the terms “*Penicillium citrinum* Fermentation product” and “Butylparaben” were introduced and later referred to in Figure 3 by the abbreviations “PCF” and “BP”, respectively. Although these abbreviations were explained in the figure legend, I feel that it may also be helpful to define them in the original section of text where they are first introduced.

Response:

Done as suggested (Line 130 and 140).

Q7) Figure 4i: Remove the word “in” from the y-axis label.

Response:

Done as suggested (Fig. 3i).

Q8) Line 155: Change the word “invested” to “investigated”.

Response:

Done as suggested (Line 169).

Q9) Line 160: Although these experiments convincingly demonstrate that *P. citrinum* can outcompete *V. lecanii* in an *in vitro* coculture, I would exercise some caution in stating that this ability is necessarily due to its antimicrobial properties, as you did not demonstrate that directly in this assay. Properties such as relative growth rates in this medium could also result in one species outcompeting another. The *in vitro* conditions (liquid culture in a shaker incubator) could also influence the outcome of these species’ relative growth rates as compared to their potential interactions in honeydew on a plant surface.

Response:

We agree with your comments. And we have rephrased the sentences as “This suggests *P. citrinum* is well adapted for growth in honeydew where it can outcompete *A. lecanii*, due to the antibiosis that has been demonstrated previously (Fig. 2e, k) and possibly a higher growth rate of *P. citrinum* in honeydew.” (Lines 173-176).

By the way, we have arranged one paragraph to discuss the interactions between *P. citrinum* and *V. lecanii*. Please see Lines 287-296.

Q10) Figure 5: The 2,4-DTBP peaks in Figures 5A (at-leg pellets) and 5B (tomato leaves) both appear to have comparable retention times (~8.4 minutes) while the peak labeled 2,4-DTBP from *Penicillium citrinum* fermentation products (Figure 5C) appears to elute a full 1 minute earlier. Is this because the samples were prepared in different solvents (hexane vs. ethyl acetate), or is there some other aspect of the analysis that I am missing?

Response:

This is because the samples were prepared in different solvents, hexane vs. ethyl acetate.

Q11) Line 178: Since you performed inhibition experiments with both *V. lecanii* and *B. bassiana* in vitro, it would have been interesting to also include *B. bassiana* in these in vivo protection experiments.

Response:

It is a very good suggestion to learn whether *in vivo* infection of *B. bassiana* is changed if pellets are present at legs, as some strains of this fungus were reported to be pathogenic to mealybugs. Actually, we have tested the pathogenicity of *B. bassiana* strain that was used in our study, and we found it was rather low for *P. solenopsis* that have no pellets. So, we did not test further for the mealybugs with pellets. Nevertheless, our result obtained from *V. lecanii* strongly suggests that pellets could offer certain degree of protection to mealybugs (Lines 485-488).

Q12) Line 185: It might be interesting to look at another metric for plant defense activation in addition to PIN2 expression, such as the accumulation of defensive compounds in the leaves following exposure to pellet-containing or pellet-less mealybugs.

Response:

Good idea! We have mentioned this in Discussion as the part of our future research. Please see Lines 308-310.

Q13) Line 188: Add the word “to” before the words “glandular trichomes”.

Response:

Done as suggested (Line 210).

Q14) Line 211: Remove the word “The” from the beginning of the sentence.

Response:

Done as suggested.

Q15) Line 219: Secretions of potatoes are mentioned in this sentence, but potatoes were previously mentioned to not produce much secretion, and mealybugs feeding on this host plant did not form pellets. It would be nice to elaborate on this discrepancy a little more.

Response:

We're sorry for the unclear description about secretions of potatoes in our study. For the original Line 219 "Secretions of *Solanum* spp. plants can adhere to insect legs and mouthparts, e.g., in the case of leafhoppers and aphids feeding on potatoes", this “potato” actually refers to wild species of potatoes (*Solanum polyadenium* and *Solanum berthaultii*), on which pellet-like structures have been observed (Tingey and Gibson, 1978; Steffens and Walters, 1991).

In the revised version of our manuscript, the secretions/trichomes of potato have been described more clearly. On Lines 102-104, we state “...and on potato (*Solanum tuberosum*) and eggplant (*Solanum melongena*) that have fewer types of trichomes relative to tomato (Glas *et al.*, 2012; Tai *et al.*, 2014; Cho *et al.*, 2017; Jayanthi *et al.*, 2018)”, and on Lines 238-240, “...*P. solenopsis* did not form pellets at their legs on potato and eggplant (Supplementary Fig. 1), which have fewer types of glandular trichomes relative to tomato (Glas *et al.*, 2012; Cho *et al.*, 2017; Jayanthi *et al.*, 2018).”

References:

Tingey, W.M., Gibson, R.W. Feeding and mobility of the potato leafhopper (Homoptera: Cicadellidae) impaired by glandular trichomes of *Solanum berthaultii* and *S. polyadenium*. *J. Econ. Entomol.* 71, 856-858 (1978).

Steffens, J.C., Walters, D.S. Biochemical aspects of glandular trichome-mediated insect resistance in the Solanaceae. *ACS Symp. Ser.* 449, 136-149 (1991).

Q16) Line 224: Are you suggesting that mealybugs redirect these secretions from their mouthparts as well? What is the fate of secretions accumulating on the mouthparts of mealybugs feeding on tomato?

Response:

We're not suggesting that mealybugs redirect these secretions from their mouthparts, but that the secretions on host surface are partially processed well by mealybugs through formation of pellets at leg. ---In the revised version, the sentence has been deleted to avoid misunderstanding.

The secretions do not accumulate on the mouthparts of mealybugs feeding on tomato.

Q17) Line 230: I think in order to call this adaptation common, it may be important to show that it occurs on more than one host plant. It is great that you could demonstrate pellet formation for multiple mealybug species, but I think only showing it on one host plant is a potential limitation of this study.

Response:

We agree with this point and suggestion. As stated in our reply to Q4, we have addressed the importance of investigating the situation on more host plants. Though we have performed such investigations as we have not discovered ideal host plants (with dense glandular trichomes) for the corresponding mealybug species.

Q18) Line 246: Pluralize "tarsus" (tarsi).

Response:

Done as suggested (Line 250).

Q19) Line 259: Change the word "outcompletes" to outcompetes.

Response:

This sentence has been deleted.

Q20) Line 264: Add the word "in" before "in vitro".

Response:

Added, please see Line 313.

Q21) Line 274: Change the word "fungi" to "fungus".

Response:

Done as suggested (Line 320).

Q22) Line 293: Change 2rd to 2nd

Response:

Done as suggested (Line 449).

Q23) Line 493: Add the word "with" after "mealybugs".

Response:

Added, please see Line 564.

Q24) Line 494: Change "mealybugs" to "mealybug".

Response:

Done as suggested (Line 565).

Q25) Lines 503-504: Change "it probes" to "they probe"

Response:

Done as suggested (Line 572).

(The end)

REVIEWERS' COMMENTS:

Reviewer #1 (Remarks to the Author):

The authors have addressed all my comments.

Reviewer #2 (Remarks to the Author):

In their revisions, the authors have addressed all my comments and concerns.

Reviewer #3 (Remarks to the Author):

The authors have adequately addressed all of my comments, although some of the added text may need minor edits for grammar and syntax. I recommend the manuscript for publication without further comments.

REVIEWERS' COMMENTS:

Reviewer #1:

The authors have addressed all my comments.

Reviewer #2:

In their revisions, the authors have addressed all my comments and concerns.

Reviewer #3:

The authors have adequately addressed all of my comments, although some of the added text may need minor edits for grammar and syntax. I recommend the manuscript for publication without further comments.

Response:

We greatly appreciate the insightful comments on our manuscript and the opportunity to submit a revision.

We have checked grammar and syntax in this revision. Please see the parts highlighted in yellow.